# DNA replication in early mammalian embryos is patterned, predisposing lamina-associated regions to fragility

Shuangyi Xu[1,8], Ning Wang[1,8], Michael V. Zuccaro [1,2], Jeannine Gerhardt[3], Rajan Iyyappan [4], Giovanna Nascimento Scatolin [4], Zongliang Jiang [4], Timour Baslan[5], Amnon Koren [6] & Dieter Egli [1,7] ✉

DNA replication in differentiated cells follows a defined program, but when and how it is established during mammalian development is not known. Here we show using single-cell sequencing, that late replicating regions are established in association with the B compartment and the nuclear lamina from the first cell cycle after fertilization on both maternal and paternal genomes. Late replicating regions contain a relative paucity of active origins and few but long genes and low G/C content. In both bovine and mouse embryos, replication timing patterns are established prior to embryonic genome activation. Chromosome breaks, which form spontaneously in bovine embryos at sites concordant with human embryos, preferentially locate to late replicating regions. In mice, late replicating regions show enhanced fragility due to a sparsity of dormant origins that can be activated under conditions of replication stress. This pattern predisposes regions with long neuronal genes to fragility and genetic change prior to separation of soma and germ cell lineages. Our studies show that the segregation of early and late replicating regions is among the first layers of genome organization established after fertilization.

DNA replication is an essential process for cell division and development, but basic principles are largely uncharacterized in the early mammalian embryo. It was recently shown that DNA replication stress defined by low replication fork speed, replication fork stalling, and a requirement for G2 DNA synthesis, is prevalent in both mouse and human embryos. Particularly in human embryos, replication stress is associated with fork collapse and chromosome breakage. As a result, human embryos frequently incur replication-dependent DNA damage and aneuploidies[1]. Abnormal chromosome content and DNA damage acquired post-fertilization impair the developmental potential of the embryo and are thus relevant for our understanding of developmental failure. Aneuploidies acquired after fertilization are also commonly seen in porcine, rhesus macaques, and bovine embryos[2-5], while in mice spontaneous aneuploidies are uncommon[6,7]. Thus, genome instability appears to be the norm in mammals, with humans and mice at opposite ends of the spectrum for both the developmental potential of a fertilized egg and the incidence of chromosomal abnormalities. Furthermore, viable genetic change during early cell divisions may result in developmental defects in the fetus and disease in the newborn, such as through error-prone fork recovery pathways and

[1]Division of Molecular Genetics, Department of Pediatrics and Naomi Berrie Diabetes Center, Columbia Stem Cell Initiative, Columbia University, New York, NY 10032, USA. [2]Graduate Program, Department of Cellular Physiology and Biophysics, Columbia University, New York, NY, USA. [3]The Ronald O. Perelman and Claudia Cohen Center for Reproductive Medicine, Weill Cornell Medical School, New York, NY, USA. [4]Department of Animal Sciences, Genetics Institute, University of Florida, Gainesville, FL, USA. [5]Department of Biomedical Sciences, School of Veterinary Medicine, The University of Pennsylvania, Philadelphia, PA, USA. [6]Department of Molecular and Cellular Biology, Roswell Park Comprehensive Cancer Center, Buffalo, NY, USA. [7]Division of Reproductive Endocrinology and Infertility, Department of Obstetrics and Gynecology, Columbia University, New York, NY, USA. [8]These authors contributed equally: Shuangyi Xu, Ning Wang. ✉e-mail: de2220@cumc.columbia.edu

chromosomal rearrangements. This has added relevance, as DNA replication stress in human embryos was shown to primarily affect regions containing long genes implicated in neurodevelopmental disorders. The molecular determinants underlying this pattern of chromosome breakage and aneuploidy are unknown.

Replication timing, a temporal order of DNA replication, is thought to play an important role in shaping the genomes of multicellular organisms, with late replicating regions experiencing higher mutation rates and fragility[8]. In somatic cells, common fragile sites (CFS) are late replicating regions, which are prone to break upon induction of replication stress[9]. In the early mammalian embryo, replication fork speed is physiologically slow even without added aphidicolin[1,10]. Spontaneous DNA breakage occurs in the embryo at sites concordant with CFS in somatic cells[1], raising the question of whether DNA replication is also patterned in the mammalian embryo. In frog, fly, and zebrafish embryos before the midblastula transition, DNA replication is very rapid and near random, with an organized replication program emerging gradually[11–13]. Such randomness might also be expected in mammalian embryos, as key epigenetic properties of chromatin affecting DNA replication patterns are only being established: chromatin architecture, which is linked to DNA replication patterns[14], solidifies during embryonic genome activation[15], before the re-establishment of DNA methylation. However, the S-phase in mammalian embryos proceeds much slower, taking hours instead of minutes as in lower vertebrates, and the cell cycle in mammalian embryos requires between 15–24 h to complete instead of less than 1.5 h for zebrafish and frogs[1,16–18]. Therefore, when defined DNA replication patterns in mammalian embryos form cannot be inferred and has not been experimentally determined.

While in somatic cells transcription-replication conflicts contribute to replication stress, genome instability at CFS in early preimplantation-stage embryos occurs prior to and independent of embryonic transcription[1]. This early pattern of chromosome fragility indicates that DNA replication may be patterned even prior to embryonic genome activation. Possible determinants of the patterns of fragility may be limiting origin density and late DNA replication, where the consequences of low fork speed and DNA replication fork stalling and fork collapse may be greatest.

Here we use mouse 1-cell and both mouse and bovine cleavage stage embryos to determine the DNA replication timing during cleavage development. We chose 1-cell and cleavage stage embryos for analysis as these are time points before and after transcriptional activation of the embryo genome. It is also the developmental time point associated with chromosome breakage and might thus provide insight into why the genome of some mammalian embryos shows a specific pattern of fragility. We find that late replicating regions emerge already in the first cell cycle, and a replication timing program is evident at the 2-cell stage and 4-cell stage in mice, as well as in early bovine cleavage stage embryos before embryonic genome activation (EGA). Long genes expressed primarily in terminally differentiated cells such as neurons, neuronal gene clusters, as well as long intergenic regions replicate late in the cell cycle. Late replicating regions show nuclear lamina association, low origin density, and increased fragility. Thus, the early establishment of DNA replication timing in the mammalian embryo predisposes specific regions to genetic change in the soma and the germ line.

## Results

### Mouse 1-cell stage and cleavage stage embryos show patterned progression of DNA replication

To determine the replication timing profile of mammalian embryos, we analyzed mouse 1-cell stage, 2-cell stage, and 4-cell stage cleavage stage embryos, as well as parthenogenetically activated embryos containing only a maternal genome. Cleavage stage embryos were dissociated, and the genome of individual cells was amplified and

sequenced (Fig. 1A). The dataset included in the analysis encompassed 318 individual nuclei from 1-cell stage embryos, and 173 2-cell stage and 153 4-cell stage mouse blastomeres, for a total of 644 samples. Recently developed methods to measure DNA replication timing of single cells were used, based on reading frequency mapped to the respective reference genome[19]. G1 phase mouse embryonic stem cells were used as a reference to even copy numbers across the genome. Individual cells/nuclei are displayed in the order according to the percent genome replicated (Fig. 1B). To define early and late replicating regions, replication timing profiles were summarized by counting the number of cells with replicated versus un-replicated DNA at each genomic bin. Here we define late replication as regions where less than 50% of the samples are replicated at each genomic bin. DNA replication timing patterns were apparent from the 1-cell stage: maternal and paternal nuclei isolated from fertilized 1-cell stage embryo showed concordant regions of late DNA replication (Fig. 1B), as did parthenogenetic embryos with only a maternal genome (Fig. S1). At the 2-cell and the 4-cell stage of fertilized embryos (Fig. 1B) and of parthenogenetic embryos (Supplementary Fig. 1), individual blastomeres showed an ordered progression from unreplicated to replicated across the genome, with congruent areas of replicated or unreplicated DNA. Aggregate replication profiles, indicated as 'summary', demonstrate distinct early and late replicating regions throughout the genome at the 1-cell, 2-cell, and 4-cell stages (Fig. 1B and Supplementary Fig. 1).

Correlation analysis showed that 1-cell and 2-cell embryos showed similar replication patterns, and 4-cell embryos were more similar to 2-cell embryos than to 1-cell embryos (Supplementary Fig. 2A). Maternal and paternal DNA replication profiles at the 1-cell stage were closely correlated ($R = 0.85$ for all autosomes Supplementary Fig. 2A, and $R = 0.84$ for chromosome 1, Supplementary Fig. 2B). Furthermore, fertilized 4-cell embryos and parthenogenetic 4-cell embryos, which contain only a maternal genome, showed highly similar replication patterns ($R = 0.93$, Fig. 1B and Supplementary Fig. 2C), suggesting no major differences between maternal and paternal genomes.

We also made comparisons with single-cell replication timing analysis from a previous study on embryonic stem (ES) cells[20], primordial germ cells, and iPSCs[21,22]. With developmental progression from the fertilized 1-cell, 2-cell to 4-cell embryos, we found increasing correlation of replication patterns with PGCs ($R = 0.62, 0.75, 0.88$), and with mouse ES single cells ($R = 0.53, 0.67, 0.87$ for all autosomes, and $R = 0.53, 0.66, 0.85$ for chromosome 1, Supplementary Fig. 2D–F), as well as with mouse iPSCs ($R = 0.49, 0.64, 0.84$) (Fig. 1B, Supplementary Fig. 2A).

Local differences between preimplantation embryos and ESCs are apparent in the replication timing display, indicated with an arrow in Fig. 1B. Furthermore, we compared aggregate single-cell replication timing patterns with replication timing patterns of differentiated cells, including mouse embryonic fibroblasts and myoblasts[21,22]. Correlation with differentiated cells was lowest at the 1-cell stage and increased toward the 4-cell stage (Supplementary Fig. 2A). 4-cell stage embryos showed lower correlation with myoblasts ($R = 0.76$) and mouse embryonic fibroblasts ($R = 0.64$) than with PGCs ($R = 0.88$) and ESCs ($R = 0.87$) (Supplementary Fig. 2A). Thus, mammalian embryos show patterned progression of DNA replication from the first cell cycle. These patterns are changing dynamically during cleavage development, and increasingly resemble those of primordial germ cells and embryonic stem cells.

### Late DNA replication correlates with LADs, B compartment, early replication with high gene and origin density

Lamina Associated Domains (LADs) is the first chromatin architecture pattern established after fertilization, before the establishment of epigenetic patterns on histones and before DNA methylation[23]. In somatic cells, LADs are known to be correlated with gene-poor late replicating regions[24]. To determine whether embryo replication timing

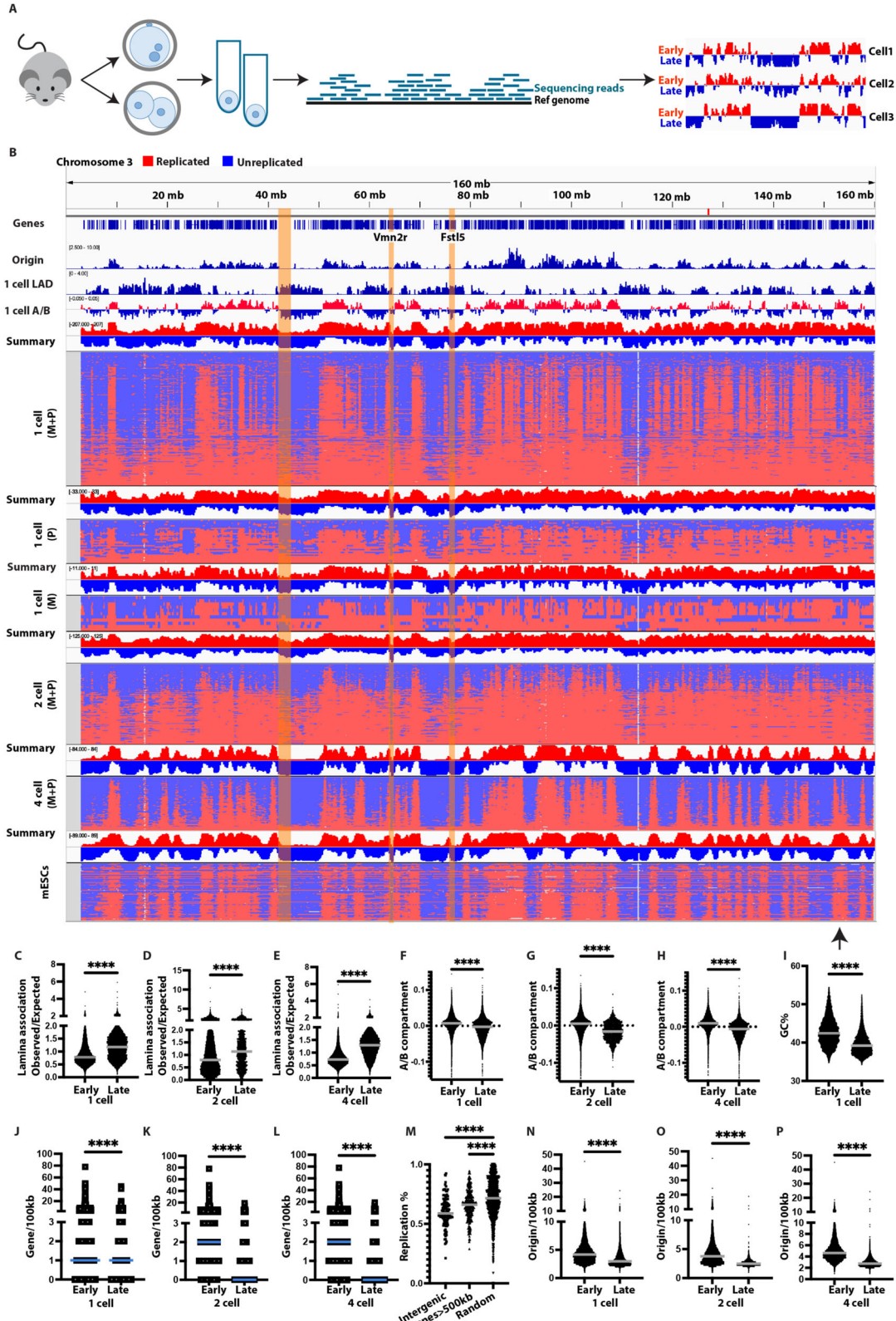

patterns correlated with LADs, we compared 1-cell stage embryo LADs with replication timing, as well as blastomere LADs with blastomere replication timing. All stages of the embryo show the visible correlation between replication timing and LADs patterns (Fig. 1B). Late replicating regions start to show greater than average (OE ratio) lamina association as early as the maternal and paternal pronuclei, continuing through development from 1-cell to blastomeres at the 2-cell and 4-cell

stages (Fig. 1C–E and Supplementary Fig. 1B–F). We also examined the correlation between replication timing and nuclear compartmentalization. In somatic cells, late replicating regions localize to the B compartment, characterized as surrounding the nucleolus and near the nuclear lamina[24]. In maternal and paternal pronuclei, as well as both fertilized and parthenogenetic embryos, late replicating regions are associated with the B-compartment (Fig. 1F–H and Supplementary

**Fig. 1 | DNA replication timing in mouse embryos is patterned, correlating with gene and origin density. A** Schematic of the experiment, including collection of individual cells from 1-cell and cleavage stage embryos, whole genome amplification, DNA sequencing, alignment of DNA reads to the genome, and replication timing for single cells. From the 1-cell stage embryos, individual haploid nuclei were isolated. Early replicating areas are expected to be overrepresented relative to late replicating areas. Blue and red colors are collapsed to one dimension in single cell replication timing profiles below. **B** DNA replication timing in mouse embryos on chromosome 3 for fertilized 1-cell embryo (1-cell M + P, *N* = 216), fertilized paternal (P) pronuclei (1-cell P, *N* = 33) and maternal (M) pronuclei (1-cell M, *N* = 11), fertilized 2-cell embryo (2-cell M + P, *N* = 117), fertilized 4-cell embryo (4-cell M + P, *N* = 82), and mouse ESCs on chromosome 3 from Dileep et al.[20], respectively. Gene density, Lamina-associated domains, and A/B compartments from 1-cell and 2-cell stage embryos[23,37], origin density from mouse embryonic stem cells[26], are shown. Vomeronasal 2 receptor (Vmn2r) gene clusters and a long neuronal gene, *FSTL5*, as well as an intergenic region are three types of late replicating regions, indicated as yellow shaded areas. **C–E** Quantification of fertilized 1-cell (**C**), 2-cell (**D**), and 4-cell stage (**E**) LADs in correlation with late replication timing. Lamina association observed/expected (OE) > 1 indicates higher lamina association than random. **F–H** Quantification of A/B compartments in early and late replicating regions in mouse fertilized 1-cell (**F**), 2-cell (**G**), and 4-cell stage (**H**) embryos. Positive value indicates A compartment and negative values indicates B compartment. **I** Quantification of GC% in early and late replicating regions in mouse fertilized 1-cell stage embryos. **J–L** Quantification of gene density in early and late replicating regions in mouse 1-cell stage embryos (**J**), 2-cell (**K**), and 4-cell embryos (**L**). **M** Quantification of replication percentage in 2-cell embryos at long genes (over 500 kb), at intergenic regions >1 Mb, and at randomly selected regions. **N–P** Quantification of origin counts in early and late replicating regions in mouse 1-cell stage embryos (**N**), 2-cell (**O**), and 4-cell embryos (**P**). Statistical test according to two-tailed Mann-Whitney test (****\**p* < 0.0001). Source data are provided as a Source Data file.

---

Fig. 1G–K). This association is highly significant from the first cell cycle (Fig. 1F), continuing through the 2-cell and 4-cell stages (Fig. 1G, H). Similarly, GC rich regions were earlier replicating than AT rich regions, consistent with their preferentially internal nuclear localization[25] (Fig 1I and Supplementary Fig. 1L–R). Thus, the link between late replication and lamina association, AT content, and the segregation of replication timing according to nuclear compartmentalization into A and B compartments, begins before the major wave of embryonic genome activation, which occurs at the 2-cell stage in mice.

To further understand the properties of replication timing in the mammalian embryo, we compared replication timing to gene density and to origin density. Late-replicating regions in mouse embryos included regions of low gene density, such as the gene *FSTL5*, a gene spanning over 600 kb and expressed in the nervous system (Fig. 1B). We quantified gene density in 100 kb bins throughout the genome and found that early-replicating regions identified in both parthenogenetic and fertilized 1-cell stage embryos and 2-cell and 4-cell cleavage stage embryos contained significantly more protein-coding genes than late-replicating regions (Fig. 1J–L and Supplementary Fig. 1S–W). Long genes with transcripts over 500 kb and in particular intergenic regions over 1 Mb were late-replicating in both 1-cell and cleavage-stage embryos, including in embryos that were entirely maternally derived (Fig. 1B, M and Supplementary Fig. 1X, Y). However, some gene-rich regions were late replicating, in particular, gene clusters expressed in neuronal cell types: Regions encoding olfactory receptor (OR) genes and vomeronasal 2 receptor (*VMN2R*) are gene-rich, but late replicating from the first cell cycle (Fig. 1B, Supplementary Fig. 3 and Supplementary Data 1). We also compared DNA replication timing in embryos to origin density, using replication origin data from mouse embryonic stem cells. We used data from mouse ES cells as these are the most closely related cell types with available origin mapping data[26], and as their DNA replication timing patterns correlate closely with mouse preimplantation embryos. Late-replicating regions showed significantly lower origin density than early-replicating regions in mouse embryos (Figs. 1B, 1N–P and Supplementary Fig. 1Z–AD). This pattern also applied to gene-rich regions that were late replicating: *OLFR* and *VMN2R* gene clusters are origin poor in mouse embryonic stem cells (Supplementary Fig. 3D). In contrast, the *HOX* gene cluster was early replicating and origin-rich (Supplementary Fig. 3C, D). Thus, late replication correlates with low origin density, which diverges from gene density at neuronal gene clusters in the embryo.

### Spontaneous chromosome break sites map to late-replicating regions in bovine embryos

To determine whether patterned DNA replication timing is also observed in other mammalian species, we chose the bovine model. We analyzed 256 bovine blastomeres harvested at the 2–7 cell stage in vitro fertilized embryos[27], and 71 blastomeres from 16 cell stage embryos, corresponding to a stage immediately before and after embryonic genome activation in bovine respectively[28]. As in mice, bovine preimplantation embryos showed patterned DNA replication timing profile before EGA (Fig. 2A), and the early and late replicating regions correlates with gene density (Fig. 2B). Yet again, the cadherin *CHD6-CHD18* gene cluster region, which is expressed in differentiated neurons, was late replicating in bovine embryos (Fig. 2A).

In somatic cells, late-replicating, gene-poor, and origin-poor regions are prone to chromosome fragility[9,29,30]. Though spontaneous chromosome breakage is rather uncommon in somatic cells, replication fork slowing using low concentrations of aphidicolin is sufficient to induce preferential breakage in these regions[31]. DNA replication fork progression in mammalian preimplantation embryos is slow even without added aphidicolin[1,10], and breakage occurs often and spontaneously. Bovine embryos reproduce the frequent spontaneous mitotic aneuploidies and chromosome breakage seen in human[5,27]. In the bovine samples analyzed, aneuploidy rates were 50% (at the 2–7 cell stage) and 26% (at the 16 cell stage), with the majority coming from chromosomal losses, a characteristic bias for mitotic aneuploidy (Fig. 2C). Mouse embryos also showed a bias towards chromosomal losses, at a far lower aneuploidy rate (ranging from ~2–5% in fertilized embryos, Fig. 2C). The chromosomal coordinates of bovine break sites had not previously been reported. To determine whether late-replicating regions are prone to fragility, we identified break sites in blastomeres of bovine cleavage stage embryos (Fig. 2D and Supplementary Data 2). We used sites of copy number transition resulting from mosaic segmental chromosome aneuploidies as a readout of the sites of chromosome breakage occurring after fertilization. We identified a total of 136 break sites, which were located preferentially, though not exclusively to gene-poor regions (Fig. 2E, F and Supplementary Data 2). As previously reported for human embryos[1], break sites were gene poor both before (2–7 cell stage) and after (16 cell stage) zygotic genome activation (Fig. 2F). A break site in the gene *DPP10* shown in Fig. 2D, is in direct concordance with a break site found in human embryos[32]. Importantly, bovine fragile sites show delayed replication relative to random sites (Fig. 2G). This suggests that late replicating regions are prone to fragility in the bovine embryo.

### Paucity of dormant origins limit adaptation to replication stress at the 1-cell stage

Unlike the frequent aneuploidies and chromosome breakage in bovine embryos (45% of all blastomeres analyzed), aneuploidies and chromosomal breakage were only found in less than 5% of analyzed mouse blastomeres. The low frequency of spontaneous breakage makes mouse embryos amenable to studies on experimentally induced stress.

In somatic cells, treatment with low concentrations of aphidicolin results in chromosome fragility due to limiting origin density in gene-poor regions of the genome[29]. We examined origin density and

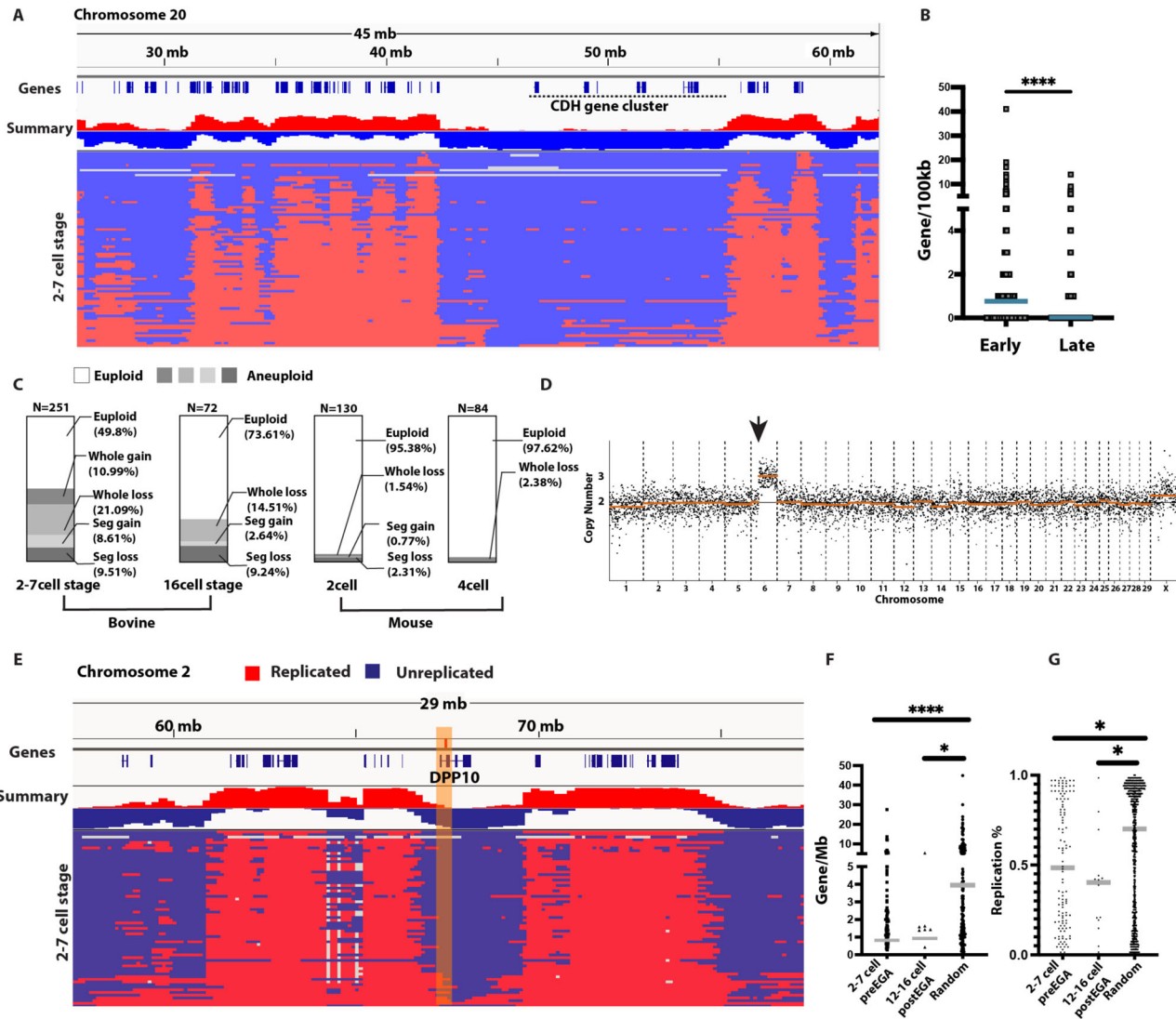

**Fig. 2 | Sites of spontaneous chromosomal breakage in bovine embryos preferentially locate to late replicating regions. A** Replication timing of bovine embryos on chromosome 20. Late replicating cadherin (CHD) gene cluster is indicated. **B** Gene density of early versus late replicating regions (****$p < 0.0001$). **C** Karyotype summary of fertilized bovine embryos and fertilized mouse embryos. Note the higher aneuploidy in bovine embryos. **D** Karyotype of a blastomere of a day 2 bovine embryo with segmental chromosomal gain. **E** DNA replication timing profile in bovine embryos on chromosome 2. The top plot shows the sum of the replication status of all cells, together with gene density. The yellow bar highlights a spontaneous chromosomal break site as reported in the long fragile site gene *DPP10*. **F** Quantification of gene density at spontaneous break sites of 2–7 cell stage embryo ($N = 122$) and 16 cell stage embryo ($N = 14$) and randomly selected regions (*$p = 0.0332$, ****$p < 0.0001$). **G** Quantification of the percentage of DNA replicated in all cells at the same spontaneous break sites identified in bovine cleavage stage embryos and at randomly selected regions (*$p = 0.0285$ for 2–7 cell preEGA, *$p = 0.0439$ for 12–16 cell postEGA). Statistical test according to two-tailed Mann-Whitney test. Source data are provided as a Source Data file.

replication fork speed using DNA fiber analysis in mouse embryos from the 1-cell stage to the blastocyst stage (Fig. 3A). We used parthenogenetic embryos with only maternal genomes, as these allow synchronous timing of cell cycle progression and precise knowledge of the timing of activation. DNA fibers were stained for IdU and for ssDNA to evaluate the continuity of the DNA fiber and origins were identified through evaluation of divergent sister forks (Fig. 3B). Median inter-origin distance was 34 kb in 1-cell embryos and increased to 81 kb at the blastocyst stage (Fig. 3C). DNA replication fork speed was evaluated by staining for both IdU and CldU (Supplementary Fig. 4). Origin density correlates inversely with replication fork speed through preimplantation development as described previously[10]. Interestingly, replication fork speed was slowest at the 1-cell stage (Fig. 3D), suggesting that the mechanisms causing fork slowing, which are currently not known, are most active during the first cell cycle. A comparison of DNA replication

fork speed between maternal only and fertilized embryos showed no appreciable differences (Supplementary Fig. 4D).

To evaluate the ability of preimplantation embryos to activate dormant origins, we incubated 1-cell embryos in low concentrations of aphidicolin during the first S-phase and determined DNA replication fork speed, origin and fork density, mitotic entry timing, and developmental potential (Fig. 3E). Aphidicolin at the 1-cell stage reduced replication fork speed in a concentration-dependent manner from 0.25 kb/min in controls to 0.21 kb/min in 0.2 µM and to 0.13 kb/min in 0.3 µM aphidicolin (Fig. 3F). Inter-origin density decreased only marginally from 34 kb to 30 kb at the 1-cell stage at 0.2 µM aphidicolin (Fig. 3G). 0.1 µM aphidicolin caused only minor, though not statistically significant changes in fork speed, origin density and fork density. Thus, the ability to activate dormant origins at the 1-cell stage is low, possibly because most available origins are already active.

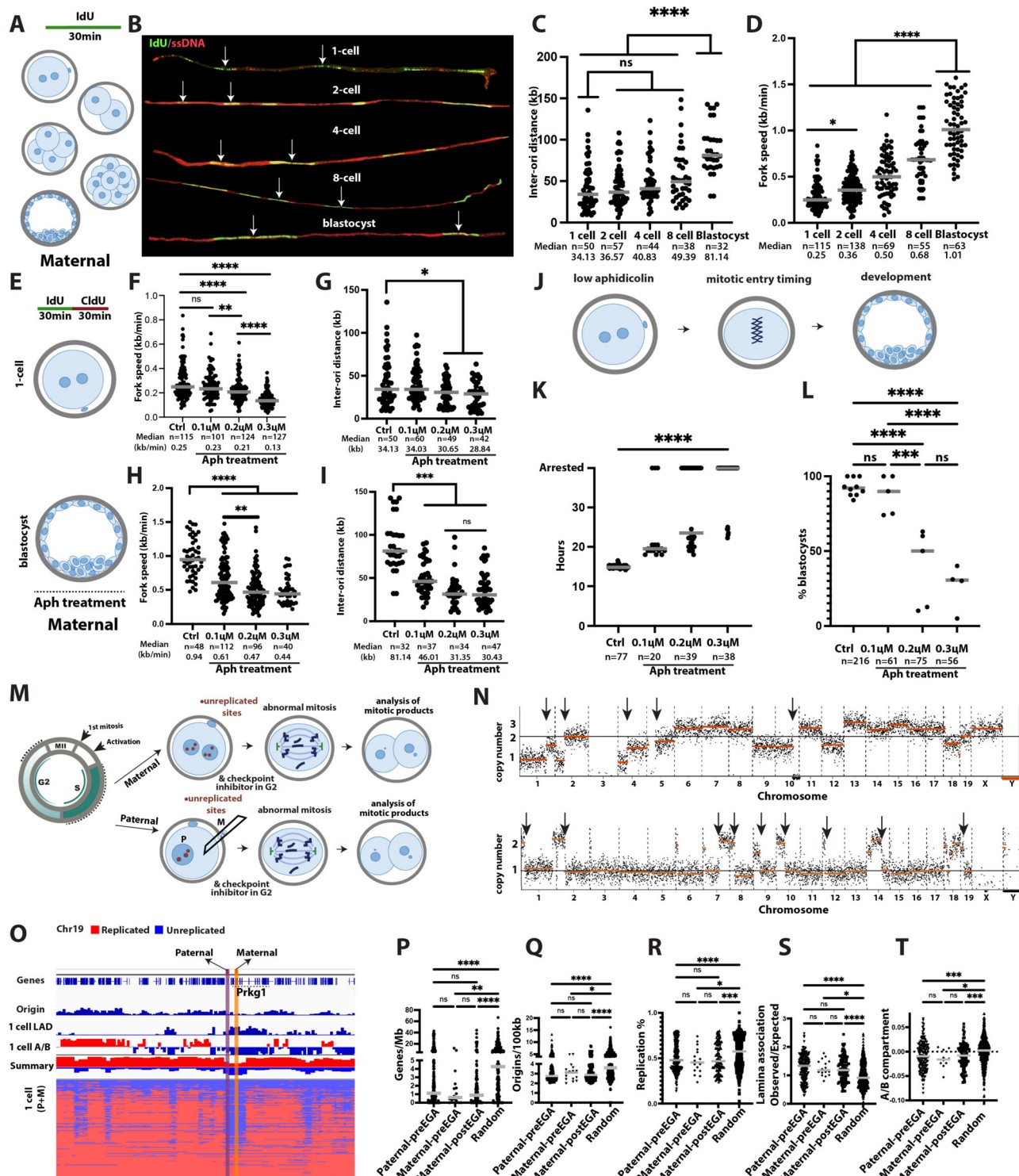

In contrast to the 1-cell stage, aphidicolin incubation at the blastocyst stage reduced replication fork speed from ~ 1 kb/min to ~ 0.5 kb/min, causing significant changes even at low concentrations of aphidicolin (Fig. 3H). Origin density increased from ~ 80 kb to 32 kb origin to origin distance, to a density observed at the 1-cell stage (Fig. 3I). This dramatic increase in replication origin density at low concentrations of aphidicolin shows a greater ability in blastocyst embryos to compensate for a reduction in fork speed and activate dormant origins than in 1-cell embryos, possibly because these origins are already physiologically active at the 1-cell stage.

Low replication fork speed at the 1-cell stage and a low ability to activate dormant origins may be limiting to DNA replication completion. To test this, we incubated mouse 1-cell embryos in low concentrations of aphidicolin (Fig. 3J), and monitored cell cycle progression 0.1 μM aphidicolin delayed entry into the first mitosis by 5 h (Fig. 3K). Thus, even minor reductions in replication fork speed have a significant effect on the kinetics of cell cycle progression. At 0.2 and 0.3 μM, mitotic entry was further delayed and most embryos failed to progress to mitosis (Fig. 3K). Though replication fork speed was reduced by only ~ 0.04 kb/min at 0.2 μM aphidicolin, most 1-cell

**Fig. 3 | Early embryos activate few dormant origins under replication stress, affecting integrity of late replicating LADs. A** Schematic of the experiment. IdU is applied for 30 min to parthenogenetic embryos. **B** Representative DNA fibers stained for IdU and ssDNA. Arrows indicate neighboring origins. Size bar = 20 kb. **C** Quantification of inter-origin distance on developmental stage. **D** Quantification of DNA replication fork speed depending on developmental stage (****$p < 0.0001$, *$p = 0.03$). **E** Schematic of the experiment. Embryos are incubated with aphidicolin and labeled with sequential pulses of IdU and CldU. **F** Quantification of DNA replication fork speed in controls and at indicated concentrations of aphidicolin at the 1-cell stage (**$p = 0.0069$). **G** Quantification of inter-origin distance at indicated conditions at 1-cell stage (*$p = 0.0399$ for ctrl-0.2uM, *$p = 0.0172$ for ctrl-0.3uM). **H** Quantification of DNA replication fork speed in controls and at indicated concentrations of aphidicolin at the blastocyst stage (***$p = 0.0002$, **$p =$ ). **I** Quantification of inter-origin distance at indicated conditions at the blastocyst stage. **J** Schematic of the experiment. Parthenogenetic mouse embryos are exposed to low concentrations of aphidicolin and cell cycle progression (**K**), and development after release from the drug (**L**) are measured. **K** Timing of mitotic entry with indicated concentrations of aphidicolin. **L** Development of mouse parthenotes with indicated concentrations of aphidicolin during the first cell cycle (***$p = 0.0002$). All statistical test above according to one-way ANOVA, ****$p < 0.0001$). **M** Schematic of the assay. Mouse embryos are incubated in low concentrations of aphidicolin throughout the S-phase. For androgenetic embryos, the maternal nucleus was removed. G2 checkpoint inhibition is applied in G2 phase to allow mitotic entry. **N** Karyotypes of mouse blastomeres with maternal (top) or paternal (bottom) genomes, exposed to low concentrations of aphidicolin at the 1-cell stage. **O** Replication timing, origin density and gene density at break site induced by low concentrations of aphidicolin. The purple vertical bar shows a paternal break site, yellow bar shows a maternal break site. **P–T** replication stress-induced break sites show lower gene density (**P**,**$p = 0.0022$), lower origin density (**Q**,*$p = 0.0358$), late replication timing (**R, ***$p = 0.0002$**), higher lamina association (**S, **$p = 0,003$**) and B compartment association (**T, ***$p = 0.0009$ for maternal postEGA, ***$p = 0.0003$ for paternal postEGA, *$p = 0.0231$**) than randomly selected sites ($N = 360$). Statistical test according to two tailed Mann-Whitney test, ****$p < 0.0001$. Source data are provided as a Source Data file.

embryos were arrested. Thus, mouse 1-cell embryos are highly sensitive to slowing of DNA replication fork speed below the already physiologically slow fork progression. Replication fork slowing due to aphidicolin exposure was damaging to developmental potential: when embryos were released from aphidicolin after 24 h of exposure, most 1-cell embryos released from 0.2 μM or 0.3 μM aphidicolin failed to develop to the blastocyst stage (Fig. 3L).

### LADs and genomic regions in B-compartment are sensitive to replication fork slowing from the first cell cycle

To identify the genomic regions with limiting origin density and long traveling forks in the early embryo, we used low concentrations of aphidicolin throughout the first S-phase and added G2 checkpoint inhibitors to facilitate mitotic entry. Upon entry into mitosis, unreplicated sites result in chromosome breakage and aneuploidy in cleavage products, which can be determined through copy number analysis (Fig. 3M). We used parthenogenetic embryos as well as androgenetic embryos for these studies to determine whether potential patterns are observed on both maternal and paternal genomes.

Low concentrations of aphidicolin resulted in abnormal chromosome content with varying copy numbers of each chromosome on either maternal or paternal genomes (Fig. 3N). The existence of the Y chromosome served as an independent confirmation of the parental origin in either group (Fig. 3N, lower panel). G2 checkpoint inhibition alone results in aneuploidy in only a third of all cells[1], and thus the vast majority of chromosomal abnormalities in this assay are caused by the slowing of replication fork progression by aphidicolin during S-phase.

Segmental errors manifest as copy number transitions within a chromosome, thereby providing the coordinates for breakage (Fig. 3N and Supplementary Data 3). For instance, Fig. 3O highlights adjacent break sites, two identified from paternal and one of maternal origin only embryos. These break sites are located at the gene *PRKG1*, a 1.2 Mb long gene, which plays a role in neurodevelopment[33]. Coordinates for a total of 140 segmental maternal and 220 paternal copy number changes were identified and gene density and origin density quantified (Supplementary Data 3). Sites of chromosome breakage on either genome were generally gene-poor (Fig. 3P) or within an *OLFR* gene cluster (Supplementary Fig. 3B), origin-poor (Fig. 3Q), and preferentially located to late-replicating regions (Fig. 3R). Furthermore, chromosome breakages were associated with 1-cell stage embryo LADs (Fig. 3S) and were enriched in the 1-cell stage embryo B compartment (Fig. 3T).

Taken together, these results show that there is a limited ability to respond to replication stress at the 1-cell stage, which is most consequential in late replicating regions that are origin-poor, gene-poor, and associated with the nuclear lamina, predisposing these regions to breakage. This pattern is established already in the first cell cycle, on both the maternal and the paternal genome. Importantly, this pattern is congruent with patterns of spontaneous chromosome breakage in cow and human embryos, and thus physiologically relevant.

### Replication gaps locate to late replicating, gene poor, origin poor regions, associated with the nuclear lamina and the B compartment from the first cell cycle

DNA replication stress results in a requirement for G2 and mitotic DNA synthesis in somatic cells[34]. Spontaneous replication stress in the embryo is associated with DNA synthesis in G2 phase, involving gaps of ~1 kb or less[1]. These are likely sites of postreplicative repair.

To evaluate the cytological location of postreplicative DNA synthesis and repair, we stained fertilized mouse 1-cell stage embryos in the late G2 phase, immediately before entry into mitosis for γH2AX, and measured proximity to the nuclear envelope and the nucleolus. Foci localized predominantly to the nuclear lamina and the nucleolus (Fig. 4A). Only 10% of all foci ($n = 331$) localized internally, away from either the nuclear envelope or the nucleolus. Of note, a greater portion of foci is associated with the nucleolus (53%) than with the nuclear lamina (37%) (Fig. 4B), adding up to 90% of foci in the B-compartment (Fig. 4C). This pattern was apparent on both maternal and paternal genomes, as identified by ~1 μm difference in nuclear diameter in G2 phase. Asymmetry was found in the number of foci, but not the pattern: the paternal genome contains significantly more foci than the maternal genome (Fig. 4C). Thus, cytological patterns of delayed replication predisposing to fragility are established during the first cell cycle, in accordance with regions of late DNA synthesis, which occurs near the nuclear lamina and around the nucleolus[35].

To determine the location of the sites requiring G2 DNA synthesis after an unperturbed first S-phase in mouse embryos, we interfered with DNA replication completion in G2 phase using high concentrations of aphidicolin combined with G2 checkpoint inhibition (Fig. 4D). Sites that remain unreplicated incur chromosome breakage in mitosis, resulting in micronucleation and aneuploidy. Genomic coordinates of segmental copy number changes caused by chromosome breakage at unreplicated sites, provide a readout for the location of unreplicated DNA in G2 phase. We used parthenogenetic mouse 1-cell embryos containing only maternal genomes for these studies, as both paternal and maternal genomes cytologically show the same patterns of postreplicative repair, and as parthenotes allow precise knowledge of cell cycle timing.

Blastomeres of the resulting 2-cell embryos showed frequent chromosomal aneuploidies with segmental errors (Fig. 4E and Supplementary Data 4). Figure 4F shows a break site in a region that is both origin-poor and gene-poor, as well as late replicating. In aggregate, sites of chromosome breakage were found to be gene-poor (Fig. 4G), origin-poor (Fig. 4H), and also replicated later than randomly selected sites at the cleavage stage (Fig.4I). In addition, breakage occurred at

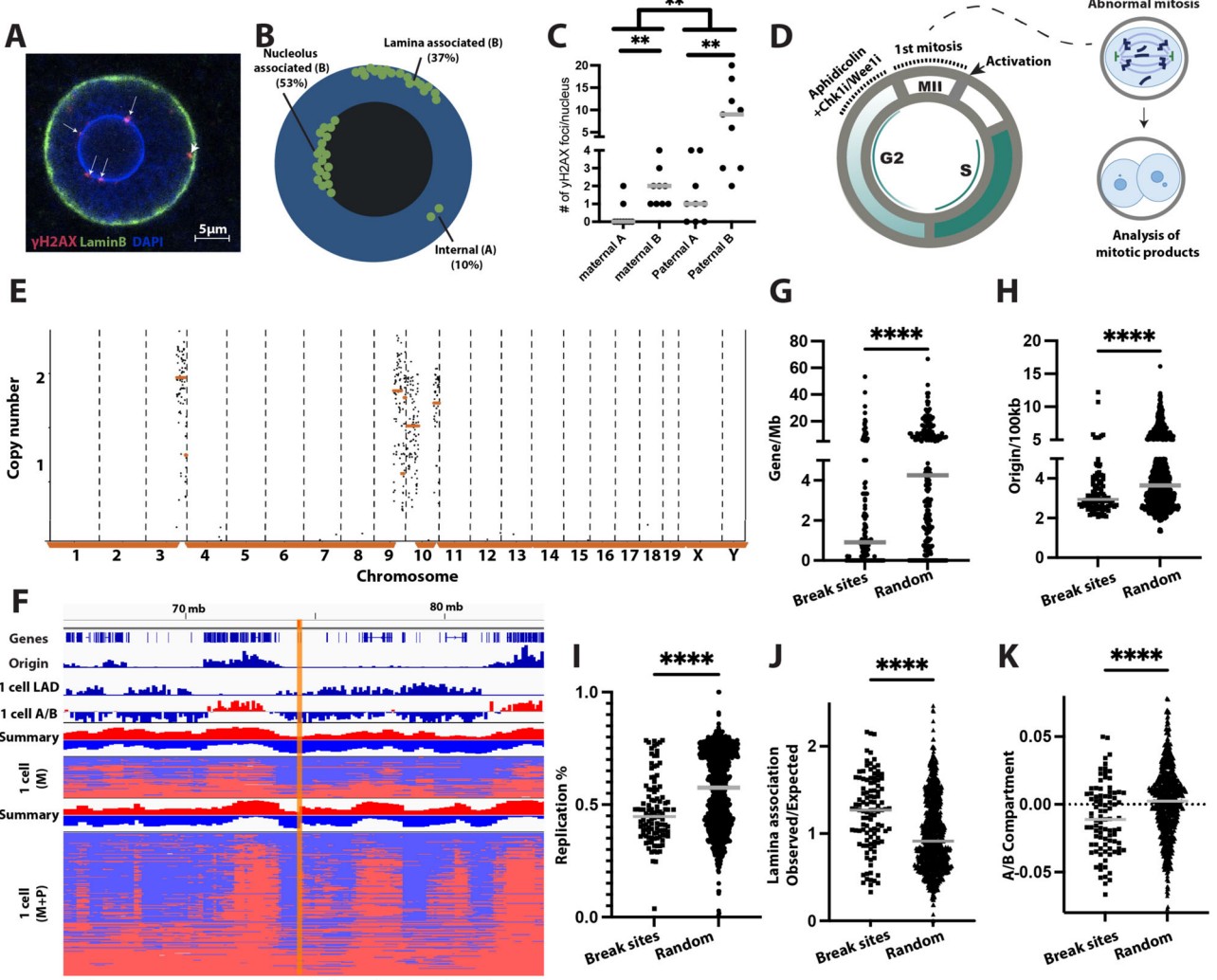

**Fig. 4 | Origin poor, lamina-associated regions show incomplete replication in late G2 phase and are prone to fragility. A** Representative immunostaining of a fertilized nucleus ($N = 18$) at late G2 phase for γH2AX and LaminB. The arrow points to foci localized on nucleolus and arrow head points to foci localized on nuclear envelope. Size bar = 5 μm. **B** Schematic of foci location. **C** Quantification of foci ($N = 115$) in different compartments in 18 nuclei; on and in proximity to nucleolus and nuclear envelope (B compartment), and A compartment. The percentage is based on A/B compartment relative to either maternal or paternal nuclei. (**$p = 0.001$). **D** Schematic of the experiment. Aphidicolin is added at high concentration at the end of the first cell cycle to inhibit DNA synthesis in late G2 phase to probe for regions with incomplete replication at that time point. CHK1 inhibition or WEE1 inhibition facilitates mitotic entry despite unreplicated DNA, resulting in chromosome breakage, and mitotic products with micronucleation and aneuploidy. Coordinates of chromosome breakage are determined using low pass single-cell genome sequencing, providing a readout for the sites of incomplete replication. **E** Chromosomal content analysis of a micronucleus isolated from a mouse blastomere after transition through the first mitosis. **F** DNA replication timing profile in mouse embryos on chromosome 6. Top shows the sum of the replication status of all cells, together with gene and origin density, LADs, and A/B compartment. The yellow vertical bar highlights the location of a chromosomal break site caused by inhibition with aphidicolin in G2 phase. **G–K** Sites of G2 replication identified through mapping of breakpoints in mice show lower gene density (**G**), lower origin density (**H**), later replication timing (**I**), higher lamina association (**J**), and B compartment association (**K**) than randomly selected sites. ($N = 111$) Statistical test according to two-tailed Mann-Whitney test (****$p < 0.0001$). Source data are provided as a Source Data file.

neuronal gene clusters, including olfactory receptor gene clusters (*OLFR*) and vomeronasal receptor gene cluster (*VMN2R*) (Supplementary Data 4). We also analyzed the correlation of break sites with LADs, showing higher association with 1-cell LADs in chromosome breakages compared with random sites (Fig. 4J). Furthermore, break sites are preferentially associated with the B compartment (Fig. 4K).

Chromosomal breakages resulting from replication fork slowing (Fig. 3) or resulting from incomplete replication in G2 phase (Fig. 4) showed concordance, affecting areas with late replication timing, low origin density, low gene density, nuclear lamina and B compartment association. Sites of direct concordance were found at *PRKG1, LRP1B, A1CF* and at two noncoding regions (Supplementary Data 5). There is also direct concordance with spontaneous fragility in other species: for

example, chromosome breakage at Lrp1b was found in untreated fertilized human embryos (Supplementary Data 5)[1]. Thus, late replicating, origin-poor, lamina-associated regions of the genome are prone to incomplete replication and to chromosome breakage. This pattern is apparent already in the first cell cycle, concordant with a temporal pattern of DNA replication progression.

## Discussion

Here we show that DNA replication in mammalian embryos progresses in a defined pattern with early replicating gene-rich regions and late-replicating, origin-poor regions containing long neuronal genes. In addition, long intergenic regions and gene clusters expressed in highly specialized neuronal cell types, including olfactory receptor genes,

and vomeronasal receptor genes and cadherin genes were also late replicating. These same regions are also late replicating in somatic cells, including in neuronal progenitors, where they are also associated with the B compartment[36]. Replication timing patterns in mice showed a high correlation with mouse embryonic stem cells, and despite the conservation of basic principles throughout development, they show a lower correlation with differentiated somatic cells. The molecular causes for these differences are not currently known, but are expected to occur in parallel to changes in the epigenome. We show that late replicating regions in the embryo have an association with the nuclear lamina and the B compartment, and have low origin density as had previously been shown for more differentiated cells[24,26]. Surprisingly, DNA replication timing in early mammalian embryos follows basic principles applicable to more differentiated cell types, despite their naïve state. This finding contrasts with the cleavage stage embryos of lower vertebrates, frog, fly and fish, which show near random DNA replication[11–13].

The early embryo provides a unique model system to study the establishment of epigenetic regulation of the genome. We find that DNA replication patterns are established already in the first cell cycle. Previous studies have shown that lamina associated domains are also becoming apparent from the 1-cell stage[23], while DNA methylation and histone methylation patterns are established only later in development. Similarly, compartments of accessible and inaccessible chromatin are established as early as the 1-cell stage[37], as are patterns of chromatin architecture[38]. Though we cannot formally exclude the possibility that a small subset of cells replicates their genome randomly at the 1-cell stage, the presence of a replication pattern argues against random replication of paternal or maternal genomes in the early mammalian embryo. The maternal genome has been reported to show delayed lamina association compared to the paternal genome at the 1-cell stage, gradually assimilating to the paternal genome after the 8-cell stage[23]. Given this as well as other asymmetries in DNA methylation between maternal and paternal genomes, one might have expected major differences in DNA replication between paternal and maternal genomes. While some differences remain possible, paternal and maternal genomes show highly similar patterns of DNA replication timing, both correlating with LADs, A/B compartmentalization, gene density, and origin density. The cytological analysis is in concordance with these findings: from the 1-cell stage, both paternal and maternal genomes show organized spatial and temporal progression of DNA replication, with late replicating regions associated with the nucleolus and the nuclear lamina[35]. Adding to that, we show that both DNA damage foci in the late G2 phase, as well as very late replicating regions prone to breakage in mitosis, show a distinct pattern of nuclear lamina and nucleolar association in both maternal and paternal genomes. Studies on nuclear lamina association using Lamin-DamID may thus underestimate the compartmentalization of the genome in the early embryo, because of a very prominent nucleolus. Recent studies in cultured cells show that domains with nucleolar association also show late DNA replication timing[39]. Though these various correlations are known for more differentiated cells, the importance here is to show that the program exists before other layers of epigenetic regulation and that it sets the stage for genome stability from the beginning of development and before the segregation of soma and germ line.

The question of developmental timing when DNA replication patterns are established is relevant to the question of cause and consequence. We find that in both bovine and mouse embryos, replication timing patterns form before embryonic genome activation. Though the experimental systems differ in their preparation (bovine eggs are in vitro matured while mouse eggs are matured in vivo), this suggests that early establishment of DNA replication timing is a conserved property in mammals. Taken together, the data presented here show that the segregation of early and late replicating DNA are among the earliest epigenetic features installed on the mammalian genome

during development and that both the nuclear lamina and the nucleolus are involved in establishing these patterns.

We chose bovine cleavage stage embryos for this analysis because the genome of bovine embryos is very unstable during cleavage development. Remarkably, bovine embryos reproduce the spontaneous pattern of breakage seen in human embryos, occurring preferentially in gene-poor regions. Breakages at concordant sites between bovine and human embryos were found on *KCNMA1* and *FHOD3* locus, which are both involved in nervous system development[40,41], in *FHIT*, a fragile site found in somatic cells exposed to aphidicolin and in tumor cells[31,42] and at *DPP10*, a long neuronal gene, and a hotspot of copy number variants[43]. Fragile sites in bovine embryos preferentially, though not exclusively, locate to late-replicating regions. Also in mice, fragile sites are concordant with human embryos, both in the type of regions prone to breakage, as well as specific sites, such as *LRP1B* and *PRKG1* (Supplementary Data 5). Such conservation of fragile sites in the early mammalian embryo is an interesting conundrum, as they confer a reproductive disadvantage and are under negative selective pressure.

In early mammalian embryos, the replication program is strained from intrinsic replication stress and has a limited ability to activate additional origins in response to exogenous stress. Regions in the genome most prone to breakage show a relative paucity of origins and replicate in G2 phase. These patterns of G2 replication and fragility are established from the first cell cycle, in association with the nuclear lamina. Lamina-associated regions are known to be fragile in the germ line and in cancer cells. For instance, the genes *DMD* and *PARK2* are lamina-associated regions with structural changes observed in the germ line in patients, and in tumor cells[44]. Breakage at both of these loci has been observed in mammalian embryos (this study and Palmerola et al.[1]). Our observations raise the question of whether genetic change in the germ line arises as a result of replication stress as early as the 1-cell stage. Patterns of DNA replication and the resulting patterns of genome stability in totipotent cells may shape mammalian genome evolution, giving rise to clonal genetic change in a manner less available to lower vertebrates as these establish replication patterns only after hundreds of nuclei have been formed[13]. Future studies should examine the causal relationships of fragility patterns, DNA replication timing, germline mutations, and other layers of epigenome regulation in the early mammalian embryo.

## Methods
### Mouse embryos
Mouse oocytes were obtained from B6D2F1 females 5–8 weeks of age from Jackson laboratories (stock # 100006), after hormonal stimulation of 5 IU PMSG per mouse, followed 48 h later by 5 IU hCG per mouse. Oocytes were dissected from oviducts 14–16 h post hCG application and dissociated in hyaluronidase. In vitro fertilization (IVF) was performed using sperm from 21–20 weeks old male mice extracted from the epididymis, in Global total for fertilization (LGTF-050). Artificial activation of mouse oocytes was performed using 1 µM ionomycin in Global Total for 5 min. at 37 deg., followed by 3.5 h 10 µg/ml puromycin and 5 µg/ml cytochalasinB, followed by 1.5 h in cytochalasinB only. Mouse embryos were then cultured in Global Total (LGGT-030) at 37 deg. in 5% CO2 atmosphere and harvested 7–9 hours post activation/fertilization for the 1-cell stage, 20–23 hours post activation for 2-cell stage, and 28 h for 4-cell stage. The timing window at the 1-cell stage is critical to obtain replication timing profiles. 1-cell stage embryos and blastomeres were isolated using laser-assisted dissection in PGD medium (LPGG-020). For the 1-cell stage embryos, 462 single maternal and paternal nuclei isolated from 2PN embryos at 7–9 h were analyzed and 332 passed quality controls (described below). A call for maternal or paternal nuclei were only made when the size of the nuclei was recognizably different. Calling was vetted based on the presence of Y chromosome in the paternally called group alone.

At the timing of collection relevant to measurement of DNA replication timing, the position relative to the polar body was not found to be informative. 1-cell stage embryo collections are sensitive to timing of collection, potentially contributing to a higher dropout rate. 88 blastomeres from 4-cell stage (44–46 h) fertilized embryos were sequenced, and 82 passed quality controls. In addition, 60 and 83 parthenogenetic blastomeres from 2-cell and 4-cell embryos were analyzed using sequencing and 56 and 71 passed quality controls.

Aphidicolin incubations were done from 3 h after artificial activation for low concentrations of aphidicolin throughout the first cell cycle, and at 2 μM and beginning at 11-12 h post activation for G2 incubations. Paternal and maternal nuclei at time of enucleation of fertilized 1-cell stage embryo were distinguished based on size and reliability vetted through sequencing. All Y chromosome-containing nuclei were in the paternal group (27/189).

Immunostaining was performed for γH2AX using mouse monoclonal antibody Millipore Cat# 05-636 (1:1000 dilution) and for Lamin B1 using mouse monoclonal antibody Proteintech Cat# 66095-1-Ig (1:500 dilution). The distance of foci to the nuclear envelope or the nucleolus was measured using the Zeiss Zen program.

All animal research has been reviewed and was approved by the Columbia IACUC under protocol #: AABK5552.

## Bovine embryos
Bovine blastomere data were downloaded from the Sequence read archive (PRJNA577965[27]). Of 114 bovine blastomeres, 69 passed quality controls, all of which were cells harvested at the 2–7 cell stage. These data were used for DNA replication timing analysis and for the identification of 110 breakpoints (Supplementary Data 2). In addition, 4-cell embryos and 16-cell embryos were collected. Briefly, germinal vesicle stage oocytes (GV oocytes) were collected as cumulus-oocyte complexes aspirated from slaughterhouse ovaries. In vitro maturation was conducted using BO-IVM medium (IVF Bioscience) for 22–23 h at 38.5 °C with 6% $CO_2$ to collect MII oocytes. Cryopreserved semen from a Holstein bull with proven fertility was diluted with BO-SemenPrep medium (IVF Bioscience) and added to drops containing COCs with a final concentration of $2 \times 10^6$ spermatozoa/ml for in vitro fertilization. Gametes were co-incubated in 6% $CO_2$ in air at 38.5 °C for 18 h. Embryos were then washed and cultured in BO-IVC medium (IVF Bioscience) at 38.5 °C with 6% $CO_2$. Different developmental stage embryos (4-cell and 16-cell) were then evaluated under light microscopy and only Grade 1 embryos by standards of the International Embryo Technology Society were selected for further study. Blastomere collection was performed as described above for mice.

Additional bovine oocytes were obtained from DeSoto Biosciences for break site identification. Maturing oocytes retrieved from ovaries, and shipped in maturation medium consisting of TCM199, FSH and 10% FBS. At 37 h post maturation, bovine oocytes were activated using 5 μM ionomycin for 5 minutes, followed by 5 μg/ml Cytochalasin B plus 6-DMAP for 4.5 h. The culture was performed at 38.5 deg. at 5% CO2 in KSOMaa (MR-106-D, EMD Millipore) with 5% FCS. Parthenogenetic blastomeres were harvested at the 8–16 cell stage for the mapping of 14 (of a total of 136) breakpoints (Supplementary Data 2).

## Single-cell sequencing and library construction
Individual cells or nuclei were collected on the heated stage @ 37 deg.C (Tokai Hit) of an Olympus IX71 inverted microscope equipped with Narishige micromanipulators and a zona pellucida laser (Hamilton-Thorne). Single nuclei were isolated by lysis and dissection of different nuclei using two 20μm diameter micropipettes (Origio). Single embryo cells or nuclei were manually picked and transferred with -0.01-0.1 μl of medium into the collection tube (column of no more than 5 mm, with a 150 μm diameter of capillary) containing 9 μl of lysis buffer, prepared as a master mix of 798 μl H2O, 6 μl of

10 mg/mL Proteinase K solution (P4850, Sigma-Aldrich), and 96 μl 10X Single Cell Lysis and Fragmentation buffer (L1043, Sigma-Aldrich). Such careful collection turns out to be critical for replication timing analysis, while break site analysis is less sensitive. Single cells / single nuclei were lysed by heating 96-well plates containing single cells/single nuclei at 50 °C for 1 h, followed by incubation at 99 °C for four minutes using a PCR thermocycler. Fragmented DNA was extended by the universal primer 5′-TGTGTTGGGTGTGTTTGGKKKKKKKKKKKKNN-3′ with Klenow polymerase (New England Biolabs, M0210L). A degenerate oligo-nucleotide primed PCR (DOP-PCR) amplification protocol that allows inline indexing of WGA DNA was applied[45]. Amplified samples were processed for Illumina library sequencing preparation using NEBNext Ultra II DNA Library Prep Kit for Illumina (New England Biolabs, E7645L). NEBNext Multiplex Oligos for Illumina (96 index primers) (New England Biolabs, E6609S) were used for four amplification cycles. Quality control for the library was done using Qubit dsDNA HS Assay Kit (Invitrogen, Carlsbad, CA, USA) and library size distribution was confirmed using a 2100 Bioanalyzer DNA 1000 Kit (Agilent, Santa Clara, CA, USA). A unimodal curve centered around 300 – 500 bp was scored as a successful library preparation. Subsequently, 30 μL of each pooled library was sent for sequencing at a concentration of 20 ng/μl. DNA sequencing was performed by Genewiz using Illumina HiSeq 4000, 2 x 150 bp, targeting a coverage of ~1-2 million reads per cell/nuclei.

## Single cell copy-number inference and break sites annotation
Sequencing data were demultiplexed according to unique barcodes and subsequently aligned to the mouse/bovine genome build (mm10/bosTau8) using tool Burrows-Wheeler Aligner v 0.7.17. Analysis of DNA sequencing data was performed as previously described[1]. Briefly, the genome was partitioned into 500 kb bins and the mapped reads were sorted, indexed and counted within the genomic bins. In addition, copy number analysis was performed using R package QDNAseq v1.26.0[46] to partition the genome into 50 kb and 100 kb bins. To distinguish real copy number changes from background noise, breakpoints were annotated only at the transition points with change in copy number >1 in the segmented read count data.

## Replication timing analysis, gene density, origin density, LADs A/B compartment and GC content correlation test
The replication timing method was adopted from the scRepli-seq[47]. Briefly, the sequencing reads were aligned to the reference genome (mm10/bosTau8) with tool Burrows-Wheeler Aligner v 0.7.17, and adapter were trimmed with cutadapt v1.18[48,49]. G1 embryonic stem cells from PRJNA427668 were used as references to even copy numbers across the genome. In bovine blastomeres, G1 cells were called according to Miura et al.[47] from the dataset PRJNA577965[27]. The mapped reads were sorted, indexed and RT scores were calculated by the control G1 cells (GSE108556). G1 cells were also used to correct for GC content and whole genome amplification efficiency. Briefly, median of the 'absolute deviations from the median' (MAD) scores were used as quality control to ensure read number variability was low (<0.3) for G1 phase cells and moderate (-0.4-0.8) for S phase cells. Samples with MAD score < 0.4 are excluded from downstream replication timing analysis as they were not called as replicating cells. With the indicated time window, the dropout rate is less than 10%. If these cells were included in the analysis, it would not alter the conclusion of a replication pattern. The replication timing data were binarized as replicated versus un-replicated regions based on the replication percentage. The replication percentage was defined as the percentage of samples replicated at each 100 kb bin. Early replicating regions were defined as >50% replicated and late replicating regions were < 50% replicated.

Gene density comparison was performed by randomly selecting locations within mm10/bosTau8 genome assembly using bedtools

v2.30.0[50]. Complete transcripts and adjacent intergenic areas were included to calculate the density of protein-coding genes.

Mouse ESC origin data were obtained from GSE68347[26]. Replication timing of mouse ESCs and differentiated cells from whole genome sequencing data was inferred using TIGER[21]. Single-cell replication timing data were obtained from Takahashi et al.[19].

Mouse 1-cell and 2-cell stage embryo non-allelic LADs data were obtained from GSE112551.

Mouse 1-cell and 2-cell stage embryo A/B compartment data were obtained from GSE82185.

Mouse GC content data were obtained from UCSC genome browser (https://genome.ucsc.edu/).

For statistical analysis, we used the Mann-Whitney test and one-way Anova as indicated in Legends.

## DNA fiber analysis

We used activated mouse oocytes, cleavage stage as well as blastocyst stage embryos for analysis. Previous studies have found no difference in DNA replication fork speed whether fertilized or activated embryos were used[1]. Cleavage and blastocyst stage embryos were asynchronous in their cell cycle at incubation.

For fork speed assay, the mouse embryos were incubated with 25 mM IdU 30 min and washed twice, and then treated with 25 mM CldU for 30 min. For fork density and origin to origin distance assays, cells were only exposed to 25 µM IdU for 30 min. Using Acidic Tyrode's solution (MR-004-D) to digest zona pellucida in a 4-well dish at room temperature for 5–10 min, and then neutralized in culture medium. The cells were collected in 1-2 ml medium and then placed in PCR tubes. Adding 20 ml of fresh pre-warmed (30 °C) spreading buffer (0.5% SDS, 50 mM EDTA, 200 mM Tris pH 7.4) to lysed cells, incubated for 6–8 min at RT and then stretched them on pre-cleaned microscope slides. Slides were fixed in pre-cooled methanol: acetic acid = 3:1 for 2 min at RT and air dried at RT, incubated in 2.5 M HCl for 50 min, and rinsed 5 times with PBS, and then blocked sides with 3% BSA in PBS for 1 h. The slides were treated with anti-BrdU, anti-IdU and anti-ssDNA antibodies for 1 h, and rinsed 3 times in PBS. Incubated with secondary antibodies for 1 h, mounted with ProLong Gold Antifade, and let dry overnight at RT. The fiber tracks were imaged on a Zeiss fluorescence microscope at 63X magnification and measured using ImageJ software v1.53. The length of each track was measured manually by ImageJ software. The pixel values were converted into µm using the scale bar generated by the microscope software. The DNA fiber length was calculated as follows: $2.59 \pm 0.24$ kbp/µm according to Jackson et al.[51]. Blinding of sample identity was used for fiber analysis.

## Reporting summary

Further information on research design is available in the Nature Portfolio Reporting Summary linked to this article.

## Data availability

Mouse and bovine blastomere sequencing data generated in this study have been deposited in the NCBI Sequence Read Archive (SRA) under accession code PRJNA874697. Source data are provided with this paper.

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

## Acknowledgements

We thank Stepan Jerabek for the critical reading of the manuscript. This work was in part supported by the Binational US-Israel Science Foundation #2021278 and NIGMS 1R01GM132604 (DE). T.B. was supported by the William C. and Joyce C. O'Neil Charitable Trust, Memorial Sloan Kettering Single Cell Sequencing Initiative. A.K. was supported by NIH R35GM148071. Z.J. was supported by NIH Eunice Kennedy Shriver National Institute of Child Health and Human Development (R01HD102533 to Z.J.) and USDA National Institute of Food and Agriculture (2019-67016-29863 to Z.J.), We thank Alberto Ciccia and Angelo Taglialatela for helpful discussions and for providing microscopy resources.

## Author contributions

S.X., N.W., and D.E. designed the study. S.X. and M.Z. performed library preparations, S.X. performed data analysis, N.W. performed DNA fiber analysis. S.X., N.W., and D.E. performed embryology. R.I., G.N.S., and Z.J. generated bovine embryos and collected 4-cell and 16-cell blastomeres. AK assisted with data analysis, T.B. contributed single-cell DNA amplification reagents and expertise, and J.G. provided help with DNA fiber analysis. S.X. and D.E. wrote the manuscript with contributions from all authors.

## Competing interests

The authors declare no competing interests.
