## [Peer Review File · Nature Communications]

DNA replication in early mammalian embryos is patterned, predisposing lamina-associated regions to fragilityREVIEWER COMMENTS

Reviewer #1 (Remarks to the Author):

In this paper, Xu et al. used mouse and bovine embryos to determine the replication timing profiles at the zygote and cleavage stage during development. They report that a distinct replication program emerges as early as in the first cell cycle. They show that the late replicating regions, that contain few and long genes, are preferentially associated with lamin and importantly, are predisposed to genomic instability.

This is an important work with implications for understanding the causes and consequences of replication stress in the early mammalian embryo.

The data analyses and interpretation are sound and the conclusions are supported by the data presented.

Major comments:

Throughout the paper, it is quite unclear whether the findings are novel or if the results align with existing knowledge from other systems. For example, the correlation between late-replicating regions and low origin density has been demonstrated elsewhere. I realize it has not been shown in this system but discussing such points could clarify what is unexpected or distinct.

Second, the authors downplay quite a bit the possibility of the maternal and paternal genomes behaving differently. I think that is quite likely, particularly in the zygote stage where it has been shown for instance, that the chromatin structure is different (<https://www.nature.com/articles/nature2326> , <https://www.nature.com/articles/nature21711> and others), and that the paternal and maternal genomes do not replicate synchronously (<https://journals.biologists.com/jcs/article/110/7/889/25256/Genome-replication-in-early-mouse-embryos-follows>).

The experiments using the parthenogenic embryos provide a lot of insights and the authors offer a rationale for using this, but I believe that it would be useful to offer the counterpoint about the paternal genome being different in some aspects.

Minor comments:

1) If I understood correctly from the methods, the cells are filtered by the MAD score. Wouldn't this bias the data set against cells that are actually replicating in the zygote but where the amplitude of the difference between early and late replication regions is small (low MAD score)?

2) Do the authors think that the results from the fiber experiments showing a high density of origins in the 1-cell stage are connected to the greater stochasticity mentioned in line 82? If so, it would be useful to connect these sections.

3) Lines 43-45: Needs reference(s)

4) In Figure 3E, in the schematic depicting 1-cell, it might be useful to highlight the maternal DNA as it is the only one being investigated.

5) There is data available on chromatin structure at the stages investigated here. Do LADs and late replicating regions correlate with B (inactive) compartments?

Reviewer #2 (Remarks to the Author):

The main objective of this study was to examine DNA replication timing during the zygote to cleavage stage of mammalian preimplantation development. Using a combination of newly generated and publicly available DNA-seq data from both mouse and bovine embryos, the authors suggest that late replicating regions emerge in the first cell cycle and that a replication timing program is evident at the 2- to 4-cell stage in mice and prior to embryonic genome activation (EGA) in cattle (i.e., 2- to 7-cell stage). They also indicate that "long genes" (over 500 Kb) that are normally expressed in terminally differentiated cells such as neurons replicate late in the cell cycle, as well as neuronal gene clusters and long intergenic regions. These late replicating regions show increased fragility due to nuclear lamina association and low replication origin density. For comparison, the authors also include parthenogenic embryos from both species and publicly available data from mouse primordial germ cells (PGC), embryonic stem cells (ESCs), induced pluripotent stem cells (iPSCs) and mouse embryonic fibroblasts (MEFs), concluding that the early establishment of DNA replication timing in the mammalian embryo predisposes specific regions to genetic change in the soma and the germ line. Overall, the manuscript is fairly well written but there are several major issues that question the novelty and validity of the authors' conclusions from the study as follows:

(1) There is a strong positive correlation between gene length, transcript length, and protein size and recent human genome studies showed that bigger transcripts are often associated with neuronal development, while smaller transcripts tend to play roles in skin and the immune system (see Lopes et al. *Front Genet* 2021 and several others). Thus, it has already been suggested that longer genes tend to be associated with functions that are important in early developmental stages, while smaller genes tend to play a role in functions that are important throughout the lifespan. This questions the overall novelty of the study.

(2) The authors state that: "Regions containing long genes are intrinsically gene-poor, contributing to the low gene density of late replicating regions," which is not surprising. Are there also differences based on chromosome size or other genomic features (GC content, etc.) that could explain and/or reinforce this?

(3) The authors show that replication timing patterns in mouse embryos are more correlated with mESCs than differentiated somatic cells and suggest that this is due to early embryos undergoing cleavage divisions and genomic instability in the absence of transcription. However, the authors state that a replication timing program is evident at the 2- to 4-cell stage in mice when EGA has already largely occurred. These statements are contradictory. Because the authors examined bovine embryos at the 2- to 7-cell stage prior to EGA, which occurs at the 8- to 16-cell stage in cattle, this could be said for the cow but would require a comparison to bovine ESCs (Bogliotti et al., *PNAS* 2018).

(4) The authors use publicly available bovine blastomere data from a previous study for comparison to mouse blastomeres generated in the current study. However, it appears this study used bovine oocytes that were matured in vitro, which is known to increase genomic instability by itself, while the current study used superovulated mouse oocytes that were matured in vivo with hCG. Therefore, these two datasets are not directly comparable.

(5) In Figure 2C, copy number is shown for a single blastomere as a representative example but there are many chromosomes exhibiting slightly less or more than 2 copies above or below the line, which questions the accuracy of the pipeline that the authors are using to call copy number variation.

(6) Along the same lines, the authors state that: "Metaphase II oocytes were used as reference to even copy numbers across the genome." However, MII oocytes can be aneuploid due to abnormal extrusion of the first polar body and thus, a known euploid somatic cell control should be used for normalization instead.

(7) Beyond having only maternal contribution to embryo development and allowing synchronous cell cycle timing, it's not entirely clear what the authors were hoping to accomplish by including data from

parthenogenetic embryos. Could intracytoplasmic sperm injection (ICSI) be used instead of parthenogenetic activation or IVF? Did the authors observe differences in replication timing with androgenetic embryos?

(8) The authors state that: "chromatin architecture, which is linked to DNA replication patterns, solidifies during embryonic genome activation, alongside major epigenetic changes in DNA methylation and histone modifications. To my knowledge, cleavage stage embryos are hypomethylated and it is not until post-implantation development that DNA methylation is re-established, which begins at the blastocyst stage in a lineage-specific manner dependent on the species.

(9) Why were only reciprocal DNA breaks examined in the study? Non-reciprocal DNA breaks should also be assessed in both mouse and bovine embryos given their relatively high frequency in the latter.

(10) Although the authors acknowledge the question of cause versus consequence from findings of G2 replication and fragility in the first cell cycle and nuclear lamina association and DNA replication timing not being fully apparent until the 2-cell stage in the Discussion section, they still conclude that early versus late replicating regions are among the first layer of epigenetic regulation established in the mammalian genome after fertilization. Because single nuclei isolated from fertilized zygotes also exhibited late and early replicating regions, albeit with lower amplitudes than cleavage-stage embryos, were there differences in late and early replicating regions between paternal and maternal pronuclei? Could this be explained by differences in H3K27me3-mediated non-conical genomic imprinting that could impact expression from the maternally and paternally derived alleles?

(11) In the Introduction section, I have no idea what the authors are trying to say by stating: "low replication fork speed replication fork stalling."

(12) The authors state that mouse spontaneous aneuploidies are uncommon and reference a previous study that performed a kinetochore counting assay, which is subjective, rather than examining copy number variation directly.

(13) The authors go back and forth between 1-cell stage and zygote, which is confusing. Pick one and stick with it.

(14) There are multiple examples of 2-sentence paragraphs that should be expanded upon or merged with another paragraph.

Reviewer #3 (Remarks to the Author):

DNA replication program has been defined in multiple kinds of cells in vitro, however its basic principle is still largely unknown in the early mammalian embryo. In this manuscript, Xu et al. characterized the replication program in bovine and mouse cleavage stage embryos using single cell sequencing, and further investigate the relationship between replication timing and genomic fragility. They found that mouse preimplantation embryos show patterned DNA replication program, with DNA replication timing patterns emerge from the zygote stage, but become apparent in 2-cell stage. Consistent with observation in somatic cells, late DNA replication correlates with LADs for both zygotes and blastomeres, while early replication correlates with high gene and origin density. In addition, it is shown that spontaneous chromosome break sites and aphidicolin-induced breaks during embryo development preferentially locate at late-replicating regions, suggesting that late replicating regions are prone to fragility. In general, the authors conclude that the pattern of DNA replication timing is among the earliest epigenetic marks established during mammalian development, and highly related with genome stability and developmental potential of the embryo.

One major concern about this paper is the novelty. It's the first time to characterize replication program in early mammalian embryo, and the authors did extensive analysis about the replication timing and genome fragility in both mice and bovine, but the concepts generated by this manuscript is more or less reported in other cells. The novelty of similar observation in early mammalian embryo should be further considered.

Here are some minor concerns

1. About the sample collection: The cell cycle duration for 2-cell and 4-cell stages is quite long, thus it should be specified how long after fertilization the embryos were chosen.
2. In Figure 1B, whether there is difference between maternal and paternal replication timing? Since there are dramatic epigenetic changes during early embryo development, and maternal and paternal DNA undergo different epigenetic reprogramming, it will be very interesting to compare their replication timing difference in zygote, 2 cells, and maybe also 4 cells. The authors discussed about this point, and claims that paternal and maternal genomes show consistent behavior in DNA replication (line 285-293), but there is no further experimental data. Is it possible to supplement this with parthenogenetic embryos for comparison to illustrate this issue.
3. In line 88-100, the authors show the replication timing of cleavage embryo closely related with embryonic stem cells. Whether there is any difference between cleavage embryo and embryonic stem cells? It will be interesting to analyze and discuss about the potential difference.
4. In Line 122, It's interesting to identify that some gene-rich regions were late replicating. What's the potential cause to make this region late replicated? Whether it is cleavage embryo specific? Is it also late replicated in neuron cells? It may be good to discuss about it in the discussion.
5. In Line 161-167, parthenogenetic embryos are used to investigate the DNA replication kinetics in 1-cell stage. In what level can the parthenogenetic embryos mimic the normal fertilized zygotes, especially at the 1-cell stage?

Reviewer #1 (Remarks to the Author)

In this paper, Xu et al. used mouse and bovine embryos to determine the replication timing profiles at the zygote and cleavage stage during development. They report that a distinct replication program emerges as early as in the first cell cycle. They show that the late replicating regions, that contain few and long genes, are preferentially associated with lamin and importantly, are predisposed to genomic instability.

This is an important work with implications for understanding the causes and consequences of replication stress in the early mammalian embryo.

The data analyses and interpretation are sound and the conclusions are supported by the data presented.

Response: we thank the reviewer for these encouraging and detailed comments and the helpful input to greatly improve the paper.

Major comments:

Throughout the paper, it is quite unclear whether the findings are novel or if the results align with existing knowledge from other systems. For example, the correlation between late-replicating regions and low origin density has been demonstrated elsewhere. I realize it has not been shown in this system but discussing such points could clarify what is unexpected or distinct.

Response: We now cite a paper showing correlation of late DNA replication with low origin density in the discussion. Late replicating regions are correlated with origin poor regions in mouse ES cells¹.

The novelty of our study is in how early this correlation is established, even as the basic principles of DNA replication have already been uncovered in other cell types. Our findings are not anticipated. We cite other studies in lower vertebrates and *Drosophila*, which show absence of discernible replication patterns during the first few replication cycles². We have now added new data which strengthen the conclusions that in mammals, DNA replication patterns are established at the 1-cell stage (**Fig. 1B**, see below). This shows that DNA replication patterns emerge prior to the major wave of embryonic gene expression concordant with the emergence of A/B compartments.

While this article was under consideration, another study reported that DNA replication timing patterns in mice emerge only after embryonic genome activation and after nuclear genome organization is established³. This passed the authors and the reviewer's scrutiny, likely because it met expectations. However, we believe this result is the consequence of the timing of sample collection and/or the quality of the data. We find, and specify this in the methods, that the collection of samples with very small media volumes is important to result in high quality DNA replication timing measurements. Also, the paper (Nakatani et al.)³ had a small number of samples in mid S phase (7-9h), and in addition, there was little control over the timing of

fertilization. We strongly believe that this paper³ came to inverse conclusions regarding the principles how DNA replication timing profiles are established, stating that the “difference in RT between the A and B compartments depends on RNA polymerase II at zygotic genome activation”. Using three different assays (replication timing in two species Fig.1&2, sites of G2 DNA synthesis Fig. 4, and chromosome fragility Fig. 3) we show that replication patterns are established in the first cell cycle and precede the major wave of embryonic genome activation. What this difference between the two studies emphasizes is just how unexpected and novel our observations are.

Second, the authors downplay quite a bit the possibility of the maternal and paternal genomes behaving differently. I think that is quite likely, particularly in the zygote stage where it has been shown for instance, that the chromatin structure is different (<https://www.nature.com/articles/nature2326> , <https://www.nature.com/articles/nature21711> and others), and that the paternal and maternal genomes do not replicate synchronously (<https://journals.biologists.com/jcs/article/110/7/889/25256/Genome-replication-in-early-mouse-embryos-follows>).

The experiments using the parthenogenetic embryos provide a lot of insights and the authors offer a rationale for using this, but I believe that it would be useful to offer the counterpoint about the paternal genome being different in some aspects.

Response: We thank you for initiating the discussion about the differences in maternal and paternal replication timing patterns. We have incorporated a new dataset and explore three aspects of maternal-paternal differences.

1. Maternal/paternal replication timing pattern. In **Fig. 1B**, we now have included replication timing profiles of maternal and paternal pronuclei. Notably, we did not observe a significant

difference between them. Correlation tests revealed a high correlation between maternal and paternal replication patterns ($R=0.85$ **Fig. S2A**). Please also note the summary diagram, which illustrates high concordance.

(We'd like to point out that discerning maternal and paternal genomes at the relevant stage is technically very challenging, because proximity to the polar body is not informative, and size differences are small. We verified the calling by analysis of sex chromosome content, which were only found in the paternal group.) We complement this by analysis of maternal only samples, as shown in **Fig. S1A**. This is significant, because previous reports showed that the establishment of a nuclear structure (LADs and 3D architecture) on the maternal genome lags behind the paternal genome, and thus if there were no pattern, it would be most apparent on the maternal genome. We find that a pattern is clearly discernible on the maternal genome (Fig. S1 below).

2. Analysis of maternal/paternal DNA damage foci γ H2AX. We stained for laminB and γ H2AX to identify and quantify spontaneous DNA damage markers in late G2 phase of the first cell cycle, when DNA methylation patterns are asymmetric (**Fig. 4A**). Our findings indicate that DNA damage markers are preferentially located in the B compartment in both maternal and paternal pronuclei (**Fig. 4C**). The only asymmetry we find is that the paternal B compartment harbors a higher abundance of foci compared to the maternal B compartment (consistent with asynchrony in replication), but no difference in the pattern was observed. In the same Figure we show that these sites of DNA repair localize to late replicating regions (**Fig. 4I**).

Figure 4A and 4C (above). Immunostaining and quantification of foci with regard to nucleolar or nuclear lamina association (B compartment).

3. Maternal/paternal chromosomal break sites.

To investigate the maternal and paternal differences in DNA replication, we created parthenogenetic and androgenetic embryos (**Fig. 3M**). Our analysis revealed that for both paternal and maternal embryos, regions limiting to DNA replication completion exposed with low concentrations of aphidicolin, exhibit identical characteristics on maternal and paternal genomes. We also find concordant sites of breakage, such as at gene *Irp1b* and *prkg1*, which were also found in spontaneous break sites in human embryo. These sites prone to breakage on either genome include low gene and origin density, late replication timing, and higher association with the nuclear lamina and the B compartment (**Fig. 3P-T**). Please also note that the pattern does not change after embryonic genome activation.

Panels of Figure 3 (above) showing breakage of maternal and paternal nucleus at the same late replicating region adjacent to the gene *Prkg1* (vertical purple and yellow lines).

Furthermore, and lastly, we compared DNA replication fork speed in maternal parthenotes and fertilized embryos, finding no differences (Fig. S4D).

Minor comments:

1) If I understood correctly from the methods, the cells are filtered by the MAD score. Wouldn't this bias the data set against cells that are actually replicating in the zygote but where the amplitude of the difference between early and late replication regions is small (low MAD score)?

Response: We no longer see low amplitude of differences in zygotes. The previous data are now improved with additional samples in the critical window of collection (7-9h). Zygotes have a very pronounced DNA replication timing profile (See Fig. 1B). We find that using optimal methods of collection, 90% of samples will successfully pass the MAD score filter. Even if those remaining 10% of samples were included, it would not alter the conclusion that there is a replication pattern.

Nevertheless, the reviewer is correct that there may be replicating cells or patterns that we do not detect in a subset of cells (<10%). We therefore modified the discussion: "we cannot formally exclude the possibility that a subset of cells replicates their genome randomly at the 1-cell stage. However, the presence of a replication pattern argues against random replication in the early embryo."

We modified the method section accordingly "Samples with MAD score <0.4 are excluded from downstream replication timing analysis as they were not called as replicating cells. At the indicated time window, the dropout rate is less than 10%. If these cells were included in the analysis, it would not alter the conclusion of a replication pattern."

2) Do the authors think that the results from the fiber experiments showing a high density of origins in the 1-cell stage are connected to the greater stochasticity mentioned in line 82? If so, it would be useful to connect these sections.

Response: We no longer see the greater stochasticity, which was based on a small dataset of samples collected at a suboptimal time point in 1-cell embryos. With a newly added dataset, we found a very pronounced replication timing pattern in zygotes. Higher origin density in zygotes is not inconsistent with a DNA replication timing pattern, see **Fig. 1B** (shown above). Zygotes intrinsically recruit a greater portion of available origins than at later stages (see **Fig. 4G** zygotes vs. blastocysts in **Fig. 4I**), but the underlying patterns remains.

3) Lines 43-45: Needs reference(s)

Response: We now cite paper on mammalian embryo cell cycle and lower vertebrates⁴⁻⁷.

4) In Figure 3E, in the schematic depicting 1-cell, it might be useful to highlight the maternal DNA as it is the only one being investigated.

Response: We now specified sample identity in the **Fig. 3A-L** and in figure legends as maternal. We now also add a comparison of replication fork speed in fertilized embryos versus maternal only embryos and find no difference (**Fig. S4D**).

The remainder of **Fig.3** investigates both paternal and maternal genomes separately (**Fig. 3M, 3P-Q**). Our findings reveal no differences in the regions prone to breakage.

5) There is data available on chromatin structure at the stages investigated here. Do LADs and late replicating regions correlate with B (inactive) compartments?

Response: This is an excellent comment, which motivated us to compare A/B compartment data with DNA replication timing. These new data show a strikingly close correlation, as now shown in **Fig. 1B, Fig. S1** and quantified in **Fig. 1F-H**.

From Figure 1 (above)

Furthermore, we also show that both sites limiting to DNA replication completion from maternal/paternal only embryos (**Fig. 3S, 3T**), as well as sites of G2 post replicative repair (**Fig. 4J, 4K**), are significantly associated with LADs and B compartments. (data source: Du et al.⁸). This was truly an excellent comment which greatly improved this manuscript.

Reviewer #2 (Remarks to the Author):

The main objective of this study was to examine DNA replication timing during the zygote to cleavage stage of mammalian preimplantation development. Using a combination of newly generated and publicly available DNA-seq data from both mouse and bovine embryos, the authors suggest that late replicating regions emerge in the first cell cycle and that a replication timing program is evident at the 2- to 4-cell stage in mice and prior to embryonic genome activation (EGA) in cattle (i.e., 2- to 7-cell stage). They also indicate that “long genes” (over 500 Kb) that are normally expressed in terminally differentiated cells such as neurons replicate late in the cell cycle, as well as neuronal gene clusters and long intergenic regions. These late replicating regions show increased fragility due to nuclear lamina association and low replication origin density. For comparison, the authors also include parthenogenic embryos from both species and publicly available data from mouse primordial germ cells (PGC), embryonic stem cells (ESCs), induced pluripotent stem cells (iPSCs) and mouse embryonic fibroblasts (MEFs), concluding that the early establishment of DNA replication timing in the mammalian embryo predisposes specific regions to genetic change in the soma and the germ line. Overall, the manuscript is fairly well written but there are several major issues that question the novelty and validity of the authors’ conclusions from the study as follows:

Response: We thank the reviewer for the constructive input on the paper, which helped us much to strengthen the conclusions, in particular regarding the early establishment of DNA replication patterns, which is the primary novel - and we believe not anticipated - finding of this study.

(1) There is a strong positive correlation between gene length, transcript length, and protein size and recent human genome studies showed that bigger transcripts are often associated with neuronal development, while smaller transcripts tend to play roles in skin and the immune system (see Lopes et al. Front Genet 2021 and several others). Thus, it has already been suggested that longer genes tend to be associated with functions that are important in early developmental stages, while smaller genes tend to play a role in functions that are important throughout the lifespan. This questions the overall novelty of the study.

Response: Despite these correlations themselves being known, this paper puts them in perspective of early replication timing development based on novel data that has been elusive for the field for many years. We believe our findings are unexpected: DNA replication timing patterns are established during the first cell cycle of development, prior to embryonic genome activation, on a largely demethylated genome. We compare these patterns to a number of genomic features: A/B domains, LAD association, origin density, gene density, as well as long genes and intergenic regions (**Fig. 1B**). We further emphasized these points in the introduction and the discussion to clarify novelty.

(2) The authors state that: “Regions containing long genes are intrinsically gene-poor, contributing to the low gene density of late replicating regions,” which is not surprising. Are there

also differences based on chromosome size or other genomic features (GC content, etc.) that could explain and/or reinforce this?

Response: Thank you for pointing out this unclear sentence, which we deleted.

Based on this comment, we performed a new analysis on GC content; we appreciate this point. We now show that late replicating regions have significantly lower GC content at fertilized 1-cell stage embryo (**Fig. 1I**), and all other stages analyzed, and whether the embryos contained both paternal and maternal genomes, or only a maternal genome (**Fig. S1L-R**). Given this early link between the two, GC content may be a driving factor in establishing DNA replication timing patterns. (we'd like to add that these patterns are not due to a bias in amplification, as it is controlled for with G1 cells).

Figure 1I Shown is GC content of early and of late replicating regions at the 1-cell stage. (As discussed above, this is a further characterization and validation of a known correlation, in a novel context.)

(3) The authors show that replication timing patterns in mouse embryos are more correlated with mESCs than differentiated somatic cells and suggest that this is due to early embryos undergoing cleavage divisions and genomic instability in the absence of transcription. However, the authors state that a replication timing program is evident at the 2- to 4-cell stage in mice when EGA has already largely occurred. These statements are contradictory. Because the authors examined bovine embryos at the 2- to 7-cell stage prior to EGA, which occurs at the 8- to 16-cell stage in cattle, this could be said for the cow but would require a comparison to bovine ESCs (Bogliotti et al., PNAS 2018).

Response: Thank you for this important point. To address this, we performed extensive new analyses at the zygote stage. These new data show a prominent DNA replication timing pattern in mice at the 1-cell stage, before EGA. Hence, like in bovine, there is a DNA replication timing pattern in mice before EGA (**Fig. 1B**, see below).

We do not have bovine ESCs available, and Bogliotti or other studies do not provide whole genome sequencing data that could be used to measure replication timing.

We were able to add additional bovine data, to enable comparison of break points at the 2-7-cell and the 16-cell stage in bovine, corresponding to before and after EGA (**Fig. 2E-F**). We found similar characteristics of spontaneous chromosomal break points, showing a bias towards late

replication and low gene density. Patterns of fragility before and after EGA likewise do not differ in mice (**Fig. 3R**).

(4) The authors use publicly available bovine blastomere data from a previous study for comparison to mouse blastomeres generated in the current study. However, it appears this study used bovine oocytes that were matured *in vitro*, which is known to increase genomic instability by itself, while the current study used superovulated mouse oocytes that were matured *in vivo* with hCG. Therefore, these two datasets are not directly comparable.

Response: This is an important point which we now acknowledge in the discussion section. We now write: "Though the experimental systems differ in their preparation (bovine eggs are *in vitro* matured while mouse eggs are matured *in vivo*), this suggests that early establishment of DNA replication timing is a conserved property in mammals."

(5) In Figure 2C, copy number is shown for a single blastomere as a representative example but there are many chromosomes exhibiting slightly less or more than 2 copies above or below the line, which questions the accuracy of the pipeline that the authors are using to call copy number variation.

Response: We thank the reviewer for pointing this out and now include new text on the calling of ploidy/copy number. There may be slight differences in read numbers on a chromosome even as the chromosomal copy number is the same. Different chromosomes tend to have somewhat different GC contents, which can contribute to these differences. Minor variations, of either biological or technical nature do not affect chromosomal copy number calling, as the magnitude of copy number changes is far larger, which is determined with above statistical analysis. Citing from the methods section: "To distinguish real copy number changes from noise, break points were annotated only at the transition points with change in copy number >1 in the segmented read count data."

(6) Along the same lines, the authors state that: "Metaphase II oocytes were used as reference to even copy numbers across the genome." However, MII oocytes can be aneuploid due to abnormal extrusion of the first polar body and a thus, a known euploid somatic cell control should be used for normalization instead.

Response: Thank you for this point. We re-evaluated our data with analysis using G1 embryonic stem cells. These are the data that we are now showing and we indicate this in the methods section. For the purpose of this response, we'd like to add that we saw no differences in the replication timing profiles, no matter which control was used. We previously used euploid MII oocytes. Though there is a filtration step to remove aneuploid cells in such comparisons, this was not needed, as mouse eggs, unlike human eggs, are very rarely aneuploid. However, this point is now mute, as the MII control is no longer used.

(7) Beyond having only maternal contribution to embryo development and allowing synchronous cell cycle timing, it's not entirely clear what the authors were hoping to accomplish by including data from parthenogenetic embryos. Could intracytoplasmic sperm injection (ICSI) be used instead of parthenogenetic activation or IVF? Did the authors observe differences in replication timing with androgenetic embryos?

Response: The reviewer raises an important point. In response, we generated a vast set of new data on fertilized zygotes, with individual analysis of paternal and maternal nuclei. We find DNA replication timing profiles in both maternal and paternal nuclei (**Fig.1B**). (the reason why M+P is larger than the separate tracks M and P is because the size difference is small, and only those that can be conclusively distinguished were called and verified according to Y chromosome content). We also separately analyzed maternal nuclei in parthenotes, which also show a distinct DNA replication timing pattern at the 1-cell stage (**Fig. S1**).

We pursued additional lines of experimentation to determine potential asymmetries between paternal and maternal genomes:

We compare sites that are prone to fragility on maternal and paternal genomes separately in **Fig. 3M-T**. These data show no differences in the pattern of breakage between paternal and maternal genomes.

(8) The authors state that: “chromatin architecture, which is linked to DNA replication patterns, solidifies during embryonic genome activation, alongside major epigenetic changes in DNA methylation and histone modifications. To my knowledge, cleavage stage embryos are hypomethylated and it is not until post-implantation development that DNA methylation is re-established, which begins at the blastocyst stage in a lineage-specific manner dependent on the species.

Response: thank you for this correction. We now are more accurate in our language: “chromatin architecture, which is linked to DNA replication patterns, solidifies during embryonic genome activation, before the re-establishment of DNA methylation.

(9) Why were only reciprocal DNA breaks examined in the study? Non-reciprocal DNA breaks should also be assessed in both mouse and bovine embryos given their relatively high frequency in the latter.

Response: We re-evaluated our data to include non-reciprocal DNA breaks in both mouse and bovine embryos. The conclusions are not impacted by this re-evaluation.

(10) Although the authors acknowledge the question of cause versus consequence from findings of G2 replication and fragility in the first cell cycle and nuclear lamina association and DNA replication timing not being fully apparent until the 2-cell stage in the Discussion section, they still conclude that early versus late replicating regions are among the first layer of epigenetic regulation established in the mammalian genome after fertilization. Because single nuclei isolated from fertilized zygotes also exhibited late and early replicating regions, albeit with lower amplitudes than cleavage-stage embryos, were there differences in late and early replicating regions between paternal and maternal pronuclei? Could this be explained by differences in H3K27me3-mediated non-conical genomic imprinting that could impact expression from the maternally and paternally derived alleles?

Response: We included new data which show that 1-cell embryos show pronounced DNA replication timing patterns, corroborating the conclusion that replication timing is among the first layers of epigenome regulation. We find these patterns on both maternal and paternal genomes. Though we cannot formally exclude differences, they both share very similar patterns ($R=0.85$).

(11) In the Introduction section, I have no idea what the authors are trying to say by stating: “low replication fork speed replication fork stalling.”

Response: thank you for pointing out the error. We changed the text to: low replication fork speed and replication fork stalling.

(12) The authors state that mouse spontaneous aneuploidies are uncommon and reference a previous study that performed a kinetochore counting assay, which is subjective, rather than examining copy number variation directly.

Response: We now cite different studies using both copy number and mitotic spreads^{9,10}. We now also examine our own spontaneous ploidy data, which are consistent with a low rate of aneuploidy in mice (less than 5% of all samples analyzed).

(13) The authors go back and forth between 1-cell stage and zygote, which is confusing. Pick one and stick with it.

Response: we now consistently use 1-cell and specify what genome is looked at (maternal or paternal).

(14) There are multiple examples of 2-sentence paragraphs that should be expanded upon or merged with another paragraph. Response: we reformatted the manuscript.

Reviewer #3 (Remarks to the Author):

DNA replication program has been defined in multiple kinds of cells in vitro, however its basic principle is still largely unknown in the early mammalian embryo. In this manuscript, Xu et al. characterized the replication program in bovine and mouse cleavage stage embryos using single cell sequencing, and further investigate the relationship between replication timing and genomic fragility. They found that mouse preimplantation embryos show patterned DNA replication program, with DNA replication timing patterns emerge from the zygote stage, but become apparent in 2-cell stage. Consistent with observation in somatic cells, late DNA replication correlates with LADs for both zygotes and blastomeres, while early replication correlates with high gene and origin density. In addition, it is shown that spontaneous chromosome break sites and aphidicolin-induced breaks during embryo development preferentially locate at late-replicating regions, suggesting that late replicating regions are prone to fragility. In general, the authors conclude that the pattern of DNA replication timing is among the earliest epigenetic marks established during mammalian development, and highly related with genome stability and developmental potential of the embryo.

Response: we thank the reviewer for these encouraging comments and the excellent input provided to improve the manuscript.

One major concern about this paper is the novelty. It's the first time to characterize replication program in early mammalian embryo, and the authors did extensive analysis about the replication timing and genome fragility in both mice and bovine, but the concepts generated by this manuscript is more or less reported in other cells. The novelty of similar observation in early mammalian embryo should be further considered.

Response: thank you for this point. We now clarify how the findings of this study differ from previous studies.

The various correlations are not novel by themselves, but the importance here is to show that the program exists before other layers of epigenetic regulation, and that it sets the stage for genome stability before segregation of soma and germ line.

How the early embryo replicates its DNA has been an important and open question for many years, and only now we have the technology to address that in mammalian cells, for which it's not been clear what to expect. This article has gained further importance with a recent Nature paper that came to different conclusions.

While this article was under consideration, another study reported that DNA replication timing patterns in mice emerge only after embryonic genome activation and after nuclear genome organization is established³. This passed the authors and the reviewer's scrutiny, likely because it met expectations. However, we believe this result is the consequence of the timing of collection and/or the quality of the data. We find, and specify this in the methods, that the collection of samples with very small media volumes is important to result in high quality DNA replication timing. Also, the paper (Nakatani et al.)³ had a small number of samples in mid S phase (7-9h), and in addition, there was little control over the timing of fertilization. The result

was that no replication profiles were detected at the 1-cell stage, poorly defined at the 2-cell stage and spotty even thereafter. The conclusion was that the “difference in RT between the A and B compartments depends on RNA polymerase II at zygotic genome activation”. Using three different assays (replication timing in two species Fig.1&2, sites of G2 DNA synthesis Fig. 4, and chromosome fragility Fig. 3) we show that patterns of replication and genome stability are established in the first cell cycle and precede the major wave of embryonic genome activation. What this difference between the two studies emphasizes is just how unexpected and novel our observations are.

Citing from discussion: “Though these various correlations are not novel by themselves, the importance here is to show that the program exists before other layers of epigenetic regulation, and that it sets the stage for genome stability before segregation of soma and germ line.”

“We show that late replicating regions in the embryo have association with the nuclear lamina and the B compartment and have low origin density as had previously been shown for more differentiated cells^{1,11}. Surprisingly, DNA replication timing in early mammalian embryos follows basic principles applicable to more differentiated cell types, despite their naïve state. This finding contrasts with the cleavage stage embryos of lower vertebrates, frog, fly and fish, which show near random DNA replication^{2,12,13}.

Most surprisingly, DNA replication patterns are established already at the 1-cell stage, prior to embryonic genome activation, correlating with A/B compartment, lamina association and GC content (**Fig.1B,1C, 1F, 1I**). This newly added analysis of A/B compartment and GC content adds to our understanding of the principles driving genome organization from the first cell cycle.

Correlations of early and late replicating regions with lamina association, A/B compartment and GC content as in Figure 1.

Here are some minor concerns

1. About the sample collection: The cell cycle duration for 2-cell and 4-cell stages is quite long, thus it should be specified how long after fertilization the embryos were chosen.

Response: this is an important point, as this can affect the results in particular at the 1-cell stage.

We now specify in the methods section for each sample when it was collected, and added additional technical details on the methods of collection.

Citing from method: "Mouse embryos were then cultured in Global Total (LGGT-030) at 37 deg. in 5% CO₂ atmosphere and harvested 7-9 hours post activation/fertilization for the zygote stage, 20-23 hours post activation for 2 cell stage, and 28 hours for 4 cell stage."

We also provided details on the methods of collection in the methods section that will be helpful to others for generation of high-quality data.

2. In Figure 1B, whether there is difference between maternal and paternal replication timing? Since there are dramatic epigenetic changes during early embryo development, and maternal and paternal DNA undergo different epigenetic reprogramming, it will be very interesting to compare their replication timing difference in zygote, 2 cells, and maybe also 4 cells. The authors discussed about this point, and claims that paternal and maternal genomes show consistent behavior in DNA replication (line 285-293), but there is no further experimental data. Is it possible to supplement this with parthenogenetic embryos for comparison to illustrate this issue.

Response: thank you for raising this important point. Our new data shed light on this issue, and this is also where one of the novelties of this paper lies. The timing of establishment and a comparison of paternal and maternal genomes tell us much about the relationship of different layers of genome regulation.

We now add new data and analysis on paternal and maternal genomes (**Fig. 1B, see below**). We find that paternal and maternal replication timing data at the zygote stage show high correlation ($R=0.85$, **Fig. S2A**). Major differences, as one might anticipate due to the asymmetry of DNA methylation in paternal and maternal nuclei, were not apparent. We supplement this with parthenogenetic embryos (**Fig. S1**).

Our extensive analysis of replication timing data from the zygote to the 4-cell stage reveals developmental differences between developmental stages (**Fig. S2A**), demonstrating that our technique is sufficiently sensitive to detect such differences.

Furthermore, we follow additional lines of experimentation to compare replication of paternal and maternal genomes. We now show equivalence in the pattern of breakage at late replicating regions in both maternal and paternal genomes (**Fig. 3P-T**), as well as direct concordance of break sites (**Fig. 3O**).

3. In line 88-100, the authors show the replication timing of cleavage embryo closely related with embryonic stem cells. Whether there is any difference between cleavage embryo and embryonic stem cells? It will be interesting to analyze and discuss about the potential difference.

Response: we concur with the reviewer that a comparison of embryonic stem cells and the early embryo is interesting. We point out such a difference in **Fig. 1B** (arrow). We believe that a comprehensive analysis of longitudinal changes along development and the molecular determinants driving these replication changes (e.g. DNA methylation, histone modifications, nuclear architecture), is best addressed in the context of a separate dedicated study.

4. In Line 122, It's interesting to identify that some gene-rich regions were late replicating. What's the potential cause to make this region late replicated? Whether it is cleavage embryo specific? Is it also late replicated in neuron cells? It may be good to discuss about it in the discussion.

Response: Our study points to low origin density (e.g. at the *Vmn2r* gene cluster in **Fig. 1B**, and **Fig. S3D**)

We do investigate the correlation of embryo replication timing patterns of the embryo with more differentiated cells in **Fig. S2A** (shown above). As the embryo progresses from the 1-cell stage to the 4-cell stage, it becomes more similar to ESCs and to a lesser extent also to fibroblasts and myoblasts. We do not currently have a suitable dataset to also compare them to mouse neuronal progenitor cells. We believe that determining the

molecular correlates and mechanisms, including changes in the epigenome, are best addressed in a figure study, perhaps including both mouse and human cells.

In the discussion, we call for further investigating causal relationships: “Future studies should examine the causal relationships of fragility patterns, DNA replication timing, germline mutations, and other layers of epigenome regulation in the early mammalian embryo.”

5. In Line 161-167, parthenogenetic embryos are used to investigate the DNA replication kinetics in 1-cell stage. In what level can the parthenogenetic embryos mimic the normal fertilized zygotes, especially at the 1-cell stage?

Response: in response to this comment, we now show data for paternal and maternal nuclei separately (**Fig. 3M, 3P-T**). Both paternal and maternal nuclei show identical patterns of chromosome breakage in response to replication fork slowing.

In addition, we also include fertilized and parthenogenetic 1-cell stage fork speed comparison that shows no significant differences (**Fig.S4D**).

Reference:

- 1 Cayrou, C. *et al.* The chromatin environment shapes DNA replication origin organization and defines origin classes. *Genome Res* **25**, 1873-1885 (2015). <https://doi.org:10.1101/gr.192799.115>
- 2 Blumenthal, A. B., Kriegstein, H. J. & Hogness, D. S. The units of DNA replication in *Drosophila melanogaster* chromosomes. *Cold Spring Harb Symp Quant Biol* **38**, 205-223 (1974).
- 3 Nakatani, T. *et al.* Emergence of replication timing during early mammalian development. *Nature* **625**, 401-409 (2024). <https://doi.org:10.1038/s41586-023-06872-1>
- 4 Palmerola, K. L. *et al.* Replication stress impairs chromosome segregation and preimplantation development in human embryos. *Cell* **185**, 2988-3007.e2920 (2022). <https://doi.org:10.1016/j.cell.2022.06.028>
- 5 Ferree, P. L., Deneke, V. E. & Di Talia, S. Measuring time during early embryonic development. *Semin Cell Dev Biol* **55**, 80-88 (2016). <https://doi.org:10.1016/j.semcdb.2016.03.013>
- 6 Gelens, L., Huang, K. C. & Ferrell, J. E., Jr. How Does the *Xenopus laevis* Embryonic Cell Cycle Avoid Spatial Chaos? *Cell Rep* **12**, 892-900 (2015). <https://doi.org:10.1016/j.celrep.2015.06.070>
- 7 Aoki, E. & Schultz, R. M. DNA replication in the 1-cell mouse embryo: stimulatory effect of histone acetylation. *Zygote* **7**, 165-172 (1999).
- 8 Du, Z. *et al.* Allelic reprogramming of 3D chromatin architecture during early mammalian development. *Nature* **547**, 232-235 (2017). <https://doi.org:10.1038/nature23263>
- 9 Fraser, L. R. & Maudlin, I. Analysis of aneuploidy in first-cleavage mouse embryos fertilized in vitro and in vivo. *Environ Health Perspect* **31**, 141-149 (1979). <https://doi.org:10.1289/ehp.7931141>

- 10 Bolton, H. *et al.* Mouse model of chromosome mosaicism reveals lineage-specific depletion of aneuploid cells and normal developmental potential. *Nature communications* **7**, 11165 (2016). <https://doi.org:10.1038/ncomms11165>
- 11 Marchal, C., Sima, J. & Gilbert, D. M. Control of DNA replication timing in the 3D genome. *Nat Rev Mol Cell Biol* **20**, 721-737 (2019). <https://doi.org:10.1038/s41580-019-0162-y>
- 12 Blow, J. J., Gillespie, P. J., Francis, D. & Jackson, D. A. Replication origins in *Xenopus* egg extract are 5-15 kilobases apart and are activated in clusters that fire at different times. *J Cell Biol* **152**, 15-25 (2001).
- 13 Siefert, J. C., Georgescu, C., Wren, J. D., Koren, A. & Sansam, C. L. DNA replication timing during development anticipates transcriptional programs and parallels enhancer activation. *Genome Res* **27**, 1406-1416 (2017). <https://doi.org:10.1101/gr.218602.116>

REVIEWERS' COMMENTS

Reviewer #1 (Remarks to the Author):

The revised version of this manuscript has addressed the points raised by the reviewers. I believe it is an important addition to the literature, as it provides detailed nuance on the important subject of when the replication timing program emerges in the embryo.

Reviewer #2 (Remarks to the Author):

In the revised version of their manuscript, Xu and colleagues addressed my questions and concerns by including additional data, performing further analyses, and clarifying important points to greatly improve the quality and clarity of the findings. I have no other other comments or suggestions for the authors.

We thank the reviewers for the input on the manuscript

No new reviewer comments were made, there are no responses